# Human coronavirus OC43-elicited CD4$^+$ T cells protect against SARS-CoV-2 in HLA transgenic mice

Rúbens Prince dos Santos Alves [1], Julia Timis[1], Robyn Miller[1], Kristen Valentine[1], Paolla Beatriz Almeida Pinto[1], Andrew Gonzalez[1], Jose Angel Regla-Nava[1,7], Erin Maule[1], Michael N. Nguyen [1], Norazizah Shafee [1], Sara Landeras-Bueno[1], Eduardo Olmedillas[1], Brett Laffey[2], Katarzyna Dobaczewska [2], Zbigniew Mikulski[2], Sara McArdle [2], Sarah R. Leist [3], Kenneth Kim [4], Ralph S. Baric [3,5], Erica Ollmann Saphire [1,6], Annie Elong Ngono [1] ✉ & Sujan Shresta [1] ✉

SARS-CoV-2-reactive T cells are detected in some healthy unexposed individuals. Human studies indicate these T cells could be elicited by the common cold coronavirus OC43. To directly test this assumption and define the role of OC43-elicited T cells that are cross-reactive with SARS-CoV-2, we develop a model of sequential infections with OC43 followed by SARS-CoV-2 in HLA-B*0702 and HLA-DRB1*0101 *Ifnar1*$^{-/-}$ transgenic mice. We find that OC43 infection can elicit polyfunctional CD8$^+$ and CD4$^+$ effector T cells that cross-react with SARS-CoV-2 peptides. Furthermore, pre-exposure to OC43 reduces subsequent SARS-CoV-2 infection and disease in the lung for a short-term in HLA-DRB1*0101 *Ifnar1*$^{-/-}$ transgenic mice, and a longer-term in HLA-B*0702 *Ifnar1*$^{-/-}$ transgenic mice. Depletion of CD4$^+$ T cells in HLA-DRB1*0101 *Ifnar1*$^{-/-}$ transgenic mice with prior OC43 exposure results in increased viral burden in the lung but no change in virus-induced lung damage following infection with SARS-CoV-2 (versus CD4$^+$ T cell-sufficient mice), demonstrating that the OC43-elicited SARS-CoV-2 cross-reactive T cell-mediated cross-protection against SARS-CoV-2 is partially dependent on CD4$^+$ T cells. These findings contribute to our understanding of the origin of pre-existing SARS-CoV-2-reactive T cells and their effects on SARS-CoV-2 clinical outcomes, and also carry implications for development of broadly protective betacoronavirus vaccines.

Severe acute respiratory syndrome coronavirus 2 (SARS-CoV-2) is the causative pathogen of the current coronavirus disease 2019 (COVID-19) pandemic. Despite the deployment of several effective SARS-CoV-2 vaccines, the pandemic has been sustained by the emergence of several variants of concern, including Beta (B.1.351), Delta (B.1.617.2), and Omicron (B.1.1.529), which displayed varying degrees of resistance to neutralizing antibodies acquired naturally or via vaccination[1–5]. Clinical manifestations of primary SARS-CoV-2 infection range in severity from asymptomatic or mild/moderate symptoms to respiratory failure, multiorgan dysfunction, and death[6–11]. While several factors, including age, gender, and comorbidities, are known to impact the clinical outcome of infection[10,12–17], little is known about whether and how pre-existing cross-reactive T-cell immunity influences the course of infection.

Several studies have shown that SARS-CoV-2 infection or vaccination can elicit robust CD8[+] and CD4[+] T cell responses in humans[18–26], and previously unexposed individuals also have functional CD8[+] and CD4[+] T cells with reactivity to SARS-CoV-2[27–32]. This pre-existing T-cell immunity is thought to result from prior exposure to 1 or more of 4 related human coronaviruses (HCoVs): OC43, HKU1, 229E, and NL63. While not yet been formally proven, this assertion is premised on 2 key observations. First, these four HCoVs are collectively responsible for about 15%-30% of common cold infections annually in adults[33], and approximately 50%-90% of the global population are seropositive for at least one of the four viruses[34]. Second, these 4 HCoVs share roughly 80% genomic sequence identity with SARS-CoV-2[35] (Table S1).

Pre-existing cross-reactive T cells have been associated with both protective and pathogenic immunity to SARS-CoV-2. Specifically, cross-reactive CD4[+] T cells have been associated with enhanced immune responses against SARS-CoV-2 infection and vaccination[36,37] and the development of severe COVID-19[38]. In contrast, pre-existing cross-reactive CD8[+] T cells have been associated with asymptomatic SARS-CoV-2 infection[39] and reduced COVID-19 severity and shorter disease duration[40–42]. There is precedent for pre-existing immunity leading to disparate effects on disease outcomes. For example, exposure to flaviviruses such as dengue and Zika viruses can either protect against or severely exacerbate subsequent infections with a different flavivirus or heterologous serotype[43–56]. Whether anti-flavivirus immunity is protective or pathogenic depends on multiple variables, including the combination of flaviviruses or serotypes, the source of cross-reactive immunity (antibody, CD8[+] T cells, CD4[+] T cells), and the time between primary and subsequent infections[57–61]. Thus, understanding the nature and effects of pre-existing cross-reactive immunity to SARS-CoV-2 has implications for identifying factors responsible for the disparate clinical outcomes among COVID-19 patients, and designing pan-coronavirus vaccines.

In this study, we investigated the antigen cross-reactivity of pre-existing OC43-elicited T cells and their biological roles during subsequent SARS-CoV-2 infection. Human leukocyte antigen (HLA) is a key determinant of the magnitude, breadth, and specificity of the T cell response in humans. To maintain human relevance, we employed our published transgenic HLA-B*0702 and HLA-DRB1*0101 Ifnar1[−/−] mouse models. HLA-B*0702 and HLA-DRB1*0101 are 2 of the most common human MHC class I and II alleles, being expressed by up to 18% and 13%, respectively, of some populations[62,63]. Deletion of type I interferon receptors (Ifnar1[−/−]) in these mice permits the study of immunity to viruses that cannot replicate in mice with an intact IFN response. Interestingly, several human cohort studies have observed an association between severe COVID-19 and inborn or acquired deficiency in the type I IFN pathway[64–70]. For example, autoantibodies against type I IFNs were detected in 20% of patients with severe COVID-19, suggesting that such autoantibodies may be a common source of acquired immune compromise[64]. We previously used HLA-B*0702 and HLA-DRB1*0101 Ifnar1[−/−] mice to demonstrate that CD8[+] and CD4[+] T cell responses to flaviviruses mirror those seen in humans with respect to antigen specificities, immunodominance patterns, T cell response kinetics, and the influence of pre-existing cross-reactive immunity in shaping the secondary T cell response and clinical outcomes[60,71–74]. Thus, with HLA transgenic Ifnar1[−/−] mice, we were able to directly address the questions of whether prior exposure to common cold HCoVs can be a source of cross-reactive SARS-CoV-2 immunity and, if so, how pre-existing cross-reactive immunity may influence the outcome of SARS-CoV-2 infection.

In this work, we first identified human-relevant immunodominant SARS-CoV-2 CD8[+] and CD4[+] T cell epitopes in two different contexts: immunization with DNA-based vaccines encoding SARS-CoV-2 spike (S), membrane (M), or nucleocapsid (N) proteins; and infection with SARS-CoV-2. We then established the cross-reactivity of OC43-elicited T cells to SARS-CoV-2 peptides, examined the effect of prior exposure

to OC43 on subsequent SARS-CoV-2 infection and lung disease, and determined the contribution of cross-reactive CD4[+] T cells to OC43-induced cross-protection. Our results demonstrate that prior exposure to OC43 generates cross-protective immunity that is mediated, at least in part, by CD4[+] T cells against SARS-CoV-2 infection and lung disease.

## Results

### Identification of SARS-CoV-2 epitopes recognized by T cells in DNA-vaccinated HLA-B*0702 and HLA-DRB1*0101 Ifnar1[−/−] mice

To investigate CD8[+] and CD4[+] T cell responses to SARS-CoV-2 in the context of human HLA alleles, we first identified the major predicted HLA-B*0702- and HLA-DRB1*0101-restricted T cell epitopes in SARS-CoV-2 S, N, and M proteins, which are known to be major CD8[+] and CD4[+] T cell targets in infected humans[23]. Using the Immune Epitope Database[75] to identify potentially immunogenic peptides, we selected the top 1% of SARS-CoV-2 S, M, and N peptides predicted to have high-affinity binding to HLA-B*0702 or HLA-DRB1*010, and obtained 69 class I-restricted epitopes (Table 1) and 42 class II-restricted epitopes (Table 2). Mice from both strains were vaccinated with a DNA-based vaccine encoding SARS-CoV-2 S, M, or N proteins (Figs. 1A, B) on days 0 and 14, and spleens and lungs were collected 7 days later (Fig. 1C). Splenocytes and lung leukocytes were incubated with each peptide (vs no-peptide control), and IFNγ-producing peptide-specific T cells were quantified by ELISpot.

Splenocytes from DNA-vaccinated HLA-B*0702 transgenic mice produced significantly higher levels of IFNγ in response to 13 of the 69 peptides ($S_{620-629}$, $S_{678-688}$, $S_{680-687}$, $S_{680-688}$, $S_{1056-1063}$, $N_{64-74}$, $N_{65-74}$, $N_{66-74}$, $N_{66-75}$, $N_{66-76}$, $N_{104-113}$, $N_{105-113}$, and $N_{105-114}$) compared to control cells; lung leukocytes from the same mice showed significantly elevated levels of IFNγ secretion in response to 7 of the 69 peptides ($S_{1056-1063}$, $N_{64-74}$, $N_{65-74}$, $N_{66-75}$, $N_{66-76}$, $N_{104-113}$, and $N_{105-113}$; Fig. 1D). Of the 7 peptides that induced significant IFNγ responses in both lung leukocytes and splenocytes, the highest response was to the 3 peptides spanning residues 104 to 113 of the N protein (Fig. 1D); this was confirmed by intracellular cytokine staining (ICS) (Fig. S1). In vaccinated HLA-DRB1*0101 mice, 3 of the 42 predicted peptides ($S_{315-329}$, $S_{959-973}$, and $M_{165-179}$) resulted in significant stimulation of IFNγ production by splenocytes, and 2 peptides ($S_{315-329}$ and $S_{998-1012}$) significantly stimulated a response in lung leukocytes; thus $S_{315-329}$ stimulated splenocytes and lung leukocytes (Fig. 1E). In contrast to the class I-restricted response, none of the N protein-derived peptides stimulated a significant IFNγ response in DNA-vaccinated HLA-DRB1*0101 mice. These data demonstrate that SARS-CoV-2 DNA-based vaccines elicited T-cell responses dominated by recognition of S and N protein-derived peptides in the spleen and lung of HLA-B*0702 Ifnar1[−/−] mice and by S and M protein-derived peptides in HLA-DRB1*0101 Ifnar1[−/−] mice.

### SARS-CoV-2 infection elicits effector CD8[+] and Th1-biased CD4[+] T cell responses in HLA transgenic Ifnar1[−/−] mice

To determine whether the antigen-specificities of the T cell response elicited by SARS-CoV-2 DNA vaccines and live virus are similar, we infected HLA-DRB1*0101 Ifnar1[−/−] mice with mouse-adapted SARS-CoV-2 MA10 strain[76] and HLA-B*0702 Ifnar1[−/−] mice with SARS-CoV-2 B.1.351 (Beta), which can replicate in mice without the need for adaptation[77,78]. Spleens were collected on day 8 post-infection (Fig. 2A), splenocytes were stimulated with select SARS-CoV-2 peptides, immunolabeled for cell surface markers, intracellular cytokines, and the degranulation marker CD107a, and the frequencies of activated (CD44[+] CD62L[−]) effector CD8[+] and CD4[+] T cells were quantified by flow cytometry (Fig. S2).

The CD8[+] T cell response in B.1.351-infected HLA-B*0702 Ifnar1[−/−] mice was assessed by stimulating splenocytes with the 6 most potent SARS-CoV-2-derived peptides identified by DNA vaccination ($S_{678-688}$, $S_{1056-1063}$, $N_{66-76}$, $N_{104-113}$, $M_{103-112}$, $M_{164-172}$). While the frequencies of activated IFNγ[+], IFNγ[+]/TNF[+], IFNγ[+]/TNF[+]/IL-2[+], and IFNγ[+]/CD107a[+] CD8[+]

## Table 1 | Predicted HLA-B*0702-restricted epitopes from SARS-CoV-2 S-, M-, and N-proteins

| Protein | Start | End | Length | Sequence |
|---|---|---|---|---|
| Spike | 24 | 32 | 9 | LPPAYTNSF |
| | 38 | 47 | 10 | YPDKVFRSSV |
| | 38 | 48 | 11 | YPDKVFRSSVL |
| | 56 | 65 | 10 | LPFFSNVTWF |
| | 207 | 216 | 10 | HTPINLVRDL |
| | 208 | 216 | 9 | TPINLVRDL |
| | 216 | 223 | 8 | LPQGFSAL |
| | 216 | 226 | 11 | LPQGFSALEPL |
| | 329 | 338 | 10 | FPNITNLCPF |
| | 411 | 419 | 9 | APGQTGKIA |
| | 462 | 472 | 11 | KPFERDISTEI |
| | 506 | 515 | 10 | QPYRVVVLSF |
| | 506 | 513 | 8 | QPYRVVVL |
| | 526 | 534 | 9 | GPKKSTNLV |
| | 588 | 597 | 10 | TPCSFGGVSV |
| | 620 | 629 | 10 | VPVAIHADQL |
| | 630 | 638 | 9 | TPTWRVYST |
| | 678 | 688 | 11 | TNSPRRARSVA |
| | 679 | 688 | 10 | NSPRRARSVA |
| | 680 | 688 | 9 | SPRRARSVA |
| | 680 | 689 | 10 | SPRRARSVAS |
| | 680 | 687 | 8 | SPRRARSV |
| | 683 | 692 | 10 | RARSVASQSI |
| | 691 | 699 | 9 | SIIAYTMSL |
| | 713 | 722 | 10 | AIPTNFTISV |
| | 714 | 722 | 9 | IPTNFTISV |
| | 727 | 736 | 10 | LPVSMTKTSV |
| | 811 | 821 | 11 | KPSKRSFIEDL |
| | 811 | 818 | 8 | KPSKRSFI |
| | 869 | 877 | 9 | MIAQYTSAL |
| | 1014 | 1022 | 9 | RAAEIRASA |
| | 1052 | 1061 | 10 | FPQSAPHGVV |
| | 1052 | 1062 | 11 | FPQSAPHGVVF |
| | 1052 | 1060 | 9 | FPQSAPHGV |
| | 1056 | 1063 | 8 | APHGVVFL |
| | 1089 | 1097 | 9 | FPREGVFVS |
| | 1089 | 1096 | 8 | FPREGVFV |
| | 1261 | 1270 | 10 | SEPVLKGVKL |
| Membrane | 58 | 67 | 10 | WPVTLACFVL |
| | 101 | 109 | 9 | RLFARTRSM |
| | 103 | 112 | 10 | FARTRSMWSF |
| | 122 | 129 | 8 | VPLHGTIL |
| | 131 | 138 | 8 | RPLLESEL |
| | 131 | 140 | 10 | RPLLESELVI |
| | 148 | 156 | 9 | HLRIAGHHL |
| | 164 | 172 | 9 | LPKEITVAT |
| Nucleoprotein | 12 | 21 | 10 | APRITFGGPS |
| | 41 | 50 | 10 | RPQGLPNNTA |
| | 45 | 53 | 9 | LPNNTASWF |
| | 64 | 74 | 11 | LKFPRGQGVPI |
| | 65 | 74 | 10 | KFPRGQGVPI |
| | 66 | 74 | 9 | FPRGQGVPI |
| | 66 | 75 | 10 | FPRGQGVPIN |
| | 66 | 76 | 11 | FPRGQGVPINT |

## Table 1 (continued) | Predicted HLA-B*0702-restricted epitopes from SARS-CoV-2 S-, M-, and N-proteins

| Protein | Start | End | Length | Sequence |
|---|---|---|---|---|
| | 66 | 73 | 8 | FPRGQGVP |
| | 67 | 74 | 8 | PRGQGVPI |
| | 93 | 101 | 9 | RIRGGDGKM |
| | 104 | 113 | 10 | LSPRWYFYYL |
| | 105 | 113 | 9 | SPRWYFYYL |
| | 105 | 114 | 10 | SPRWYFYYLG |
| | 149 | 158 | 10 | RNPANNAAIV |
| | 150 | 159 | 10 | NPANNAAIVL |
| | 150 | 158 | 9 | NPANNAAIV |
| | 161 | 171 | 11 | LPQGTTLPKGF |
| | 256 | 265 | 10 | KKPRQKRTAT |
| | 257 | 265 | 9 | KPRQKRTAT |
| | 257 | 267 | 11 | KPRQKRTATKA |
| | 257 | 264 | 8 | KPRQKRTA |
| | 308 | 317 | 10 | APSASAFFGM |

T cells were significantly increased in response to stimulation with epitope $N_{104\text{-}113}$ (vs control), there was no significant expansion in response to the other 5 peptides (Fig. 2B, with the gating strategy represented in Fig. S2A). Thus, vaccination with a SARS-CoV-2 N protein-encoding DNA vaccine and direct infection with B.1.351 elicited effector CD8+ T cell response in HLA-B*0702 $Ifnar1^{-/-}$ mice that were both directed against the immunodominant epitope $N_{104\text{-}113}$. This finding is in agreement with previous reports that SARS-CoV-2 $N_{105\text{-}113}$ is the immunodominant epitope in SARS-CoV-2-infected individuals expressing HLA-B*0702[25,28,79–82].

As the class II-restricted response to SARS-CoV-2 DNA vaccines was lower in magnitude than the class I-restricted response, we stimulated splenocytes from MA10-infected HLA-DRB1*0101 $Ifnar1^{-/-}$ mice with each of the 42 SARS-CoV-2 peptides predicted to be immunogenic in the context of HLA-DRB1*0101, and then analyzed the frequencies of activated CD4+ T cells producing IFNγ alone or IFNγ and TNF (Th1 cells), IL-4 (Th2 cells), and IL-17A (Th17 cells) (Fig. 2C, with the gating strategy represented in Fig. S2B). All 42 peptides increased the frequency of IFNγ-producing cells compared with unstimulated control cells, but the increase was significant only in response to 2 peptides: $S_{959\text{-}973}$ and $N_{107\text{-}121}$. Three peptides expanded multifunctional IFNγ+/TNF+ CD4+ T cells ($S_{315\text{-}329}$, $S_{512\text{-}526}$, and $N_{328\text{-}342}$). In contrast, none of the peptides stimulated CD4 + T cell production of IL-4 or IL-17A. Of note, $N_{107\text{-}121}$ encompasses most of the immunodominant $N_{104-113}$ CD8+ T cell epitope identified in both DNA-vaccinated and SARS-CoV-2-infected HLA-B*0702 $Ifnar1^{-/-}$ mice, and $S_{315\text{-}329}$ also stimulated splenocytes from the DNA-vaccinated HLA-DRB1*0101 $Ifnar1^{-/-}$mice. These findings demonstrate that the major CD4+ T cell response to primary infection with SARS-CoV-2 MA10 in HLA-DRB1*0101 $Ifnar1^{-/-}$ mice is Th1-biased, which is consistent with human studies showing that SARS-CoV-2 infection or vaccination elicits CD4+ T cells with a Th1-like phenotype[23,83,84]. Moreover, the agreement between the human studies and our findings in both DNA-vaccinated and SARS-CoV-2-infected HLA transgenic mice further validates these mouse models for investigation of human-relevant CD8+ and CD4+ T cell responses to SARS-CoV-2.

### OC43 infection elicits CD8+ T cells with cross-reactivity to SARS-CoV-2 in HLA-B*0702 $Ifnar1^{-/-}$ mice

The genomic sequence of SARS-CoV-2 N protein is 29% and 23% identical to the N protein sequences of β-coronaviruses (OC43 and HKU-1) and α-coronaviruses (NL63 and 229E), respectively[85,86]. To

**Table 2 | Predicted HLA-DRB1*0101-restricted epitopes from SARS-CoV-2 S-, M-, and N-proteins**

| Protein | Start | End | Length | Sequence |
|---|---|---|---|---|
| Spike | 4 | 18 | 15 | FLVLLPLVSSQCVNL |
| | 54 | 68 | 15 | LFLPFFSNVTWFHAI |
| | 62 | 76 | 15 | VTWFHAIHVSGTNGT |
| | 115 | 129 | 15 | QSLLIVNNATNVVIK |
| | 199 | 213 | 15 | GYFKIYSKHTPINLV |
| | 231 | 245 | 15 | IGINITRFQTLLALH |
| | 236 | 250 | 15 | TRFQTLLALHRSYLT |
| | 264 | 278 | 15 | AYYVGYLQPRTFLLK |
| | 315 | 329 | 15 | TSNFRVQPTESIVRF |
| | 344 | 358 | 15 | ATRFASVYAWNRKRI |
| | 363 | 377 | 15 | ADYSVLYNSASFSTF |
| | 512 | 526 | 15 | VLSFELLHAPATVCG |
| | 539 | 553 | 15 | VNFNFNGLTGTGVLT |
| | 692 | 706 | 15 | IIAYTMSLGAENSVA |
| | 758 | 772 | 15 | SFCTQLNRALTGIAV |
| | 853 | 867 | 15 | QKFNGLTVLPPLLTD |
| | 869 | 883 | 15 | MIAQYTSALLAGTIT |
| | 885 | 899 | 15 | GWTFGAGAALQIPFA |
| | 895 | 909 | 15 | QIPFAMQMAYRFNGI |
| | 902 | 916 | 15 | MAYRFNGIGVTQNVL |
| | 959 | 973 | 15 | LNTLVKQLSSNFGAI |
| | 967 | 981 | 15 | SSNFGAISSVLNDIL |
| | 998 | 1012 | 15 | TGRLQSLQTYVTQQL |
| | 1005 | 1019 | 15 | QTYVTQQLIRAAEIR |
| | 1010 | 1024 | 15 | QQLIRAAEIRASANL |
| | 1015 | 1029 | 15 | AAEIRASANLAATKM |
| | 1044 | 1058 | 15 | GKGYHLMSFPQSAPH |
| Membrane | 32 | 46 | 15 | ICLLQFAYANRNRFL |
| | 60 | 74 | 15 | VTLACFVLAAVYRIN |
| | 71 | 85 | 15 | YRINWITGGIAIAMA |
| | 91 | 105 | 15 | MWLSYFIASFRLFAR |
| | 98 | 112 | 15 | ASFRLFARTRSMWSF |
| | 116 | 130 | 15 | TNILLNVPLHGTILT |
| | 144 | 158 | 15 | ILRGHLRIAGHHLGR |
| | 165 | 179 | 15 | PKEITVATSRTLSYY |
| | 175 | 189 | 15 | TLSYYKLGASQRVAG |
| Nucleoprotein | 107 | 121 | 15 | RWYFYYLGTGPEAGL |
| | 129 | 143 | 15 | GIIWVATEGALNTPK |
| | 154 | 168 | 15 | NAAIVLQLPQGTTLP |
| | 303 | 317 | 15 | QIAQFAPSASAFFGM |
| | 328 | 342 | 15 | GTWLTYTGAIKLDDK |
| | 387 | 401 | 15 | KKQQTVTLLPAADLD |

investigate whether exposure to seasonal common cold HCoVs can elicit CD8+ T cells that cross-react with SARS-CoV-2, we infected HLA-B*0702 *Ifnar1*−/− mice with OC43, the most common seasonal HCoV worldwide[87,88]. We then analyzed viral load in upper and lower airway tissues on days 1, 3, and 5 post-infection (Fig. S3A), and CD8+ T cell responses to SARS-CoV-2 peptides on days 1, 3, 5, 8, 16, and 30 post-infection (Fig. 3A). While OC43 genomic RNA levels in nasal turbinates increased between days 1 and 5, levels in lung were highest on day 1 and below the level of detection by day 5 (Figure S3B). Splenocytes prepared on days 8 and 16 post-infection with OC43 were stimulated with a panel of 37 HLA-B*0702-restricted SARS-CoV-2 CD8+ T cell epitopes previously shown to stimulate human CD8+ T cells based on IFNγ-

ELISpot or ICS assays (NIAID Virus Pathogen Database and Analysis Resource; Table 3)[25,89–95]. These 37 peptides from SARS-CoV-2 M ($n = 3$), N ($n = 7$), ORF1ab ($n = 24$), ORF7 ($n = 2$), and ORF8 ($n = 1$) proteins—were selected to ensure that the analysis of OC43-elicited T cell reactivity was focused on the most human-relevant SARS-CoV-2 epitopes. The only significant activated CD8+ T response was at day 16 post-OC43 infection, and was focused on a single region in the N protein, with 9- and 12-fold expansion of $N_{104-121}$-reactive IFNγ+ and IFNγ/TNF+ CD8+ T cells, respectively (Fig. 3B, with the gating strategy represented in Fig. S2C). To validate this finding, splenocytes and lung leukocytes isolated on day 8 post-infection were stimulated with the 69-peptide panel (Table 1 and Fig. 1C). Indeed, the frequencies of IFNγ-producing splenocytes and lung leukocytes were increased by only 3 SARS-CoV-2 peptides, all-encompassing the $N_{104-113}$ epitope (Fig. 3C). Thus, exposure to OC43 elicits HLA-B*0702-restricted CD8+ T cells with reactivity to the SARS-CoV-2 $N_{104-113}$ epitope. Of note, the SARS-CoV-2 and OC43 $N_{104-113}$ sequences differ by a single amino acid residue (L**S**PRWYFYYL and L**L**PRWYFYYL, respectively), providing a basis for this cross-reactivity.

We extended this investigation by following the development of the SARS-CoV-2 $N_{104-113}$-reactive effector CD8+ T cell response in the spleen and lung for a period of 30 days following OC43 infection (Fig. 3A, D). The gating strategy is represented in Fig. S2A. Expansion of $N_{104-113}$-reactive IFNγ+ and IFNγ+/TNF+ CD8+ T cells was evident by day 8 and remained stable (IFNγ+) or increased slightly (IFNγ+/TNF+) by day 30. In contrast, polyfunctional IFNγ+/TNF+/IL-2+ cells were undetectable until day 30, at which point a small but significant expansion of SARS-CoV-2 $N_{104-113}$-reactive cells was detected in the spleen but not the lung. Finally, IFNγ+/CD107a+ CD8+ T cells exhibited a biphasic response detected in the spleen by day 8, had waned by day 16, and increased again by day 30. These data provide the first direct demonstration that the seasonal coronavirus OC43 elicits SARS-CoV-2 cross-reactive CD8+ T cells. They also show that this cross-reactive response is directed against a single immunodominant SARS-CoV-2 epitope, $N_{104-113}$, in the context of HLA-B*0702.

## OC43 infection elicits CD4+ T cells with cross-reactivity to SARS-CoV-2

To determine whether OC43 infection can also induce a CD4+ T cell response that cross-reacts with SARS-CoV-2, HLA-DRB1*0101 *Ifnar1*−/− mice were infected with OC43, and spleens and lungs harvested on days 8, 16, and 30 post-infection (Fig. 4A). Splenocytes and lung leukocytes isolated on day 8 were stimulated with the same 42-panel of SARS-CoV-2 peptides (Table 1) used to stimulate cells from DNA-vaccinated and SARS-CoV-2-infected mice (Figs. 1D and 2C, respectively). The frequencies of IFNγ-producing splenocytes were increased by 4 SARS-CoV-2 peptides ($S_{54-68}$, $S_{264-278}$, $S_{758-772}$, and $M_{32-46}$), and IFNγ-producing lung leukocytes were increased by 1 ($S_{264-278}$) (Fig. 4B). To validate the CD4+ T cell cross-reactivity in this model, splenocytes were stimulated with a panel of 37 HLA-DRB1*0101-restricted peptides derived from SARS-CoV-2 E, S, M, N, ORF1ab, ORF3a, and ORF8 proteins (Table 4) previously shown to stimulate human CD4+ T cell responses by IFNγ-ELISpot, ICS, or MHC-binding assays[18,90,96–100] (Fig. 4C). The gating strategy is represented in Fig. S2D. At day 8, IFNγ+ CD4+ T cells reactive with all 37 peptides were expanded in the spleen, although the increase was statistically significant only for cells stimulated with $M_{66-80}$ and $ORF3a_{116-130}$. At day 16, frequencies of IFNγ+ CD4+ T cells cross-reactive with SARS-CoV-2 $ORF8_{96-110}$ and $ORF8_{101-115}$ were significantly increased. In contrast, polyfunctional SARS-CoV-2 cross-reactive IFNγ+/TNF+ CD4+ T cells were significantly expanded only in response to $N_{86-100}$ and $N_{261-275}$ peptides. In addition, while $N_{86-100}$ and $ORF3a_{108-120}$ peptides stimulated IL-2 production at day 16, none of the 37 peptides stimulated IL-4 expression (Fig. S3G). Finally, at day 30, the CD4+ T cell response was decreased for all peptides. Thus, exposure to OC43 elicits a Th1-biased CD4+ T cell response that cross-reacts with

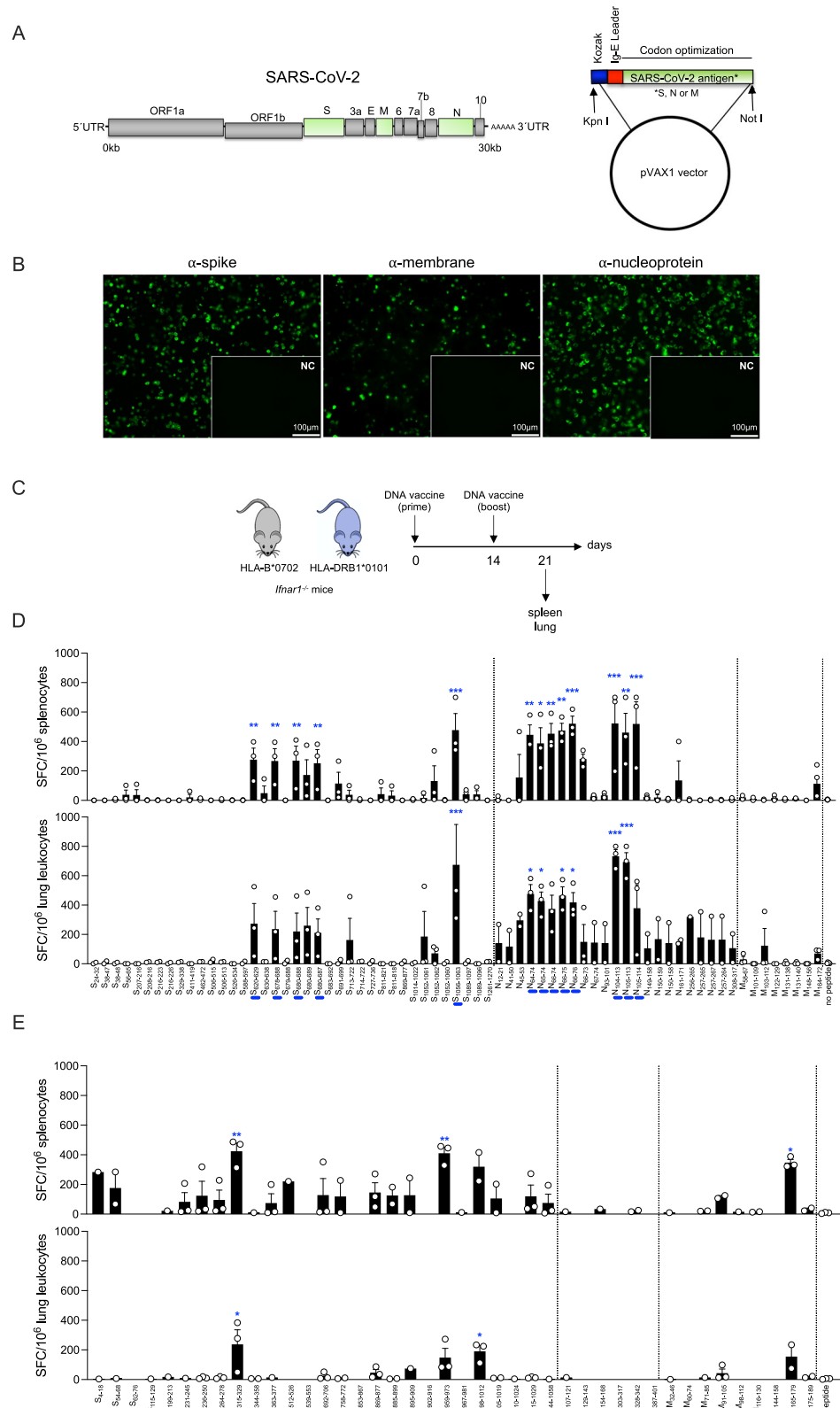

SARS-CoV-2 in the context of HLA-DRB1*0101. Viral load analysis on days 1, 3, and 5 post-infection (Fig. S3A) revealed levels of OC43 genomic RNA that were high in nasal turbinates on all 3 days and dramatically lower in lung (undetectable on day 1 and rising significantly but only slightly by day 3; Fig. S3C).

We next examined whether OC43 infection stimulates a SARS-CoV-2 cross-reactive antibody response in the context of HLA-

DRB1*0101, given that CD4[+] T cells play crucial roles in promoting and maintaining antibody responses[101,102]. To this end, serum was isolated from the OC43-infected mice at 6 time points from 0 to 100 days post-infection (Fig. S3D) and analyzed by ELISA for antibodies that bind to OC43 or SARS-CoV-2 S and N proteins. Anti-OC43 S IgG titers were detectable by day 14 and remained relatively stable up to day 100; in contrast, IgG reactive with SARS-CoV-2 S protein was not detected in

**Fig. 1 | Mapping SARS-CoV-2 S, N, and M protein-derived epitopes in DNA-vaccinated HLA-B\*0702 and HLA-DRB1\*0101 *Ifnar1*$^{-/-}$ mice. A** SARS-CoV-2 genome and DNA vaccine constructs containing mammalian-optimized Kozak sequence, IgE leader sequence, and codon-optimized DNA sequence for SARS-CoV-2 S, N, or M protein. **B** Representative immunofluorescence images of 293 T cells transfected with S, M, or N DNA vaccines (*vs* empty vector [insets]) and immuno-labeled for SARS-CoV-2 S, N, or M protein (green). Scale bars apply to main panels and insets. Images are representative of 2 independent experiments. **C** Experimental protocol for D and E. HLA-B\*0702 and HLA-DRB1\*0101 *Ifnar1*$^{-/-}$ mice were administered 25 μg S, N, or M DNA vaccines by intramuscular electroporation on days 0 and 14, and tissue collected 7 days later. **D, E** ELISpot quantification of IFNγ-producing spot-forming cells (SFC) from HLA-B\*0702 *Ifnar1*$^{-/-}$ mice (**D**) and HLA-DRB1\*0101 *Ifnar1*$^{-/-}$ mice (**E**). Splenocytes and lung leukocytes were stimulated for 20 h with 69 (**D**), or 42 (**E**) SARS-CoV-2 peptides predicted to be immunogenic for CD8$^+$ T cells (**D**) or CD4$^+$ T cells (**E**; Tables 1 and 2); control, no peptide. Data are presented as the mean ± SEM; $N = 4$ mice/group pooled from 2 independent experiments. Peptide vs control were compared using the one-way ANOVA test. The mean of each peptide was compared to the mean of the control (no peptide). *$P < 0.05$; **$P < 0.01$; ***$P < 0.001$. Blue bars on the *x*-axis are peptides that significantly stimulated 1 or more cell types.

sera of OC43-infected mice at any time point (Fig. S3E). IgG titers against the N proteins of both OC43 and SARS-CoV-2 were minimal at all time points (Fig. S3F). Thus, in our model of primary OC43 infection in HLA-DRB1\*0101 *Ifnar1*$^{-/-}$ mice, OC43 S-specific IgG and SARS-CoV-2-reactive CD4$^+$ T cells are present. These data demonstrate that prior exposure to OC43 elicits an HLA-DRB1\*0101-restricted CD4$^+$ T cell response that cross-reacts with SARS-CoV-2 epitopes.

### Immunization with N$_{104-113}$ peptide protects against SARS-CoV-2 infection and lung damage in HLA-B\*0702 *Ifnar1*$^{-/-}$ mice

To determine whether the OC43 cross-reactive CD8$^+$ T cell response can protect against or exacerbate SARS-CoV-2 infection and/or pathology, HLA-B\*0702 *Ifnar1*$^{-/-}$ mice were primed and boosted with N$_{104-113}$ peptide on days 0 and 21, challenged with SARS-CoV-2 B.1.351 at 14 days post-boost, and tissues harvested 3 days later (Fig. 5A, with the gating strategy represented in Fig. S2E). In splenocytes from N$_{104-113}$-immunized mice (*vs* mock-immunized), the frequencies of N$_{104-113}$-reactive IFNγ$^+$, polyfunctional (IFNγ$^+$/TNF$^+$ and IFNγ$^+$/TNF$^+$/IL-2$^+$) and cytotoxic multifunctional (IFNγ$^+$/CD107a$^+$) CD8$^+$ T cells were significantly increased (Fig. 5B), and lung appeared healthier by histopathology (Fig. 5C). In fact, quantification of histopathology data revealed 3 features of SARS-CoV-2-induced lung disease that were less pronounced (lower scores) in N$_{104-113}$-immunized mice, although these differences were not significant: necrosis of bronchiolar epithelial cells (BEC), cellular debris in bronchioles, and suppurative bronchiolitis. When the Fig. 5A experiment was repeated with the MA10 strain of SARS-CoV-2 (Fig. S4A), our findings were confirmed: the N$_{104-113}$-immunized mice had significantly higher frequencies of N$_{104-113}$-reactive polyfunctional CD8$^+$ T cells (Fig. S4B) and significantly lower histopathology scores (Fig. S4C). For the B.1.351-challenged mice, we also analyzed viral burden. Both RT-qPCR analysis of genomic RNA (Fig. 5D) and immunofluorescence analysis of SARS-CoV-2 N protein (Fig. 5E) revealed significantly lower levels of SARS-CoV-2 in lungs of N$_{104-113}$-immunized mice. These results demonstrate that immunization of HLA-B\*0702 *Ifnar1*$^{-/-}$ mice with SARS-CoV-2 N$_{104-113}$ peptide elicits an antigen-specific polyfunctional CD8$^+$ T cell response and protects against SARS-CoV-2 infection and lung disease.

### Prior exposure to OC43 confers cross-protection against SARS-CoV-2 infection in HLA-B\*0702 *Ifnar1*$^{-/-}$ mice

Given that N$_{104-113}$-immunization reduced SARS-CoV-2 burden and pathogenesis, we hypothesized that OC43-elicited CD8$^+$ T cell immunity might be similarly protective. To test this, HLA-B\*0702 *Ifnar1*$^{-/-}$ mice were infected with OC43 and challenged with SARS-CoV-2 on day 8 or 16 post-infection (Fig. S5A) or 60–70 days post-infection (Fig. 5F). Immunologic and virologic phenotypes were analyzed at 3 days post-challenge, which allowed us to focus on the effects of OC43-elicited immunity—rather than the primary T cell response to the SARS-CoV-2 challenge (primary antiviral T cell responses are generally not detectable until days 4 or 5 post-infection[47,48,74]).

To test for the presence of the OC43-elicited cross-reactive CD8$^+$ T cell response in HLA-B\*0702 *Ifnar1*$^{-/-}$ mice, splenocytes from OC43-exposed (*vs* naïve) SARS-CoV-2-challenged mice were stimulated with N$_{104-113}$ peptide and analyzed by ICS (Fig. 5G). Polyfunctional CD8$^+$ T cells were significantly increased in mice challenged at 60–70 days post-OC43 infection (Fig. 5G). In contrast, in mice challenged at 8 or 16 days, activated effector CD8$^+$ T cell subsets expanded, but these increases were generally insignificant (Fig. S5B). RT-qPCR analysis revealed no effect of OC43 pre-exposure on SARS-CoV-2 genomic RNA levels in either lungs or nasal turbinates of mice challenged on days 8 or 16 (Fig. S5C). In contrast, lungs from mice challenged 60 to 70 days post-OC43 infection exhibited dramatic reductions in both SARS-CoV-2 genomic RNA (Fig. 5H) and N-protein immunoreactivity (Fig. 5I). While blinded histopathological analysis of lungs revealed no differences between OC43-infected and naïve mice challenged at 8 or 16 days post-infection (Fig. S5D), OC43-infected mice challenged at 60 to 70 days tended to have more bronchioles with clear lumina and viable epithelial cells lining the airway (i.e., proper polarization), and exhibited decreases in 3 histopathologic features (necrotic epithelial cells, cellular debris within bronchioles, and bronchiolar lesions), which were, however, not significant (Fig. 5J). Thus, a single prior intranasal exposure to OC43 can protect against SARS-CoV-2 infection in HLA-B\*0702 *Ifnar1*$^{-/-}$ mice and may also limit SARS-CoV-2-induced lung damage in some mice. The immunologic and virologic data together suggest that OC43-elicited SARS-CoV-2-cross-reactive CD8$^+$ T cells may contribute to the cross-protection against SARS-CoV-2 infection.

### OC43 infection confers cross-protection against SARS-CoV-2 in HLA-DRB1\*0101 *Ifnar1*$^{-/-}$ mice in a manner partially dependent on CD4$^+$ T cells

We have shown that HLA-DRB1\*0101 *Ifnar1*$^{-/-}$ mice, mount an antigen-specific CD4$^+$ T cell response against SARS-CoV-2 after DNA vaccination or viral infection (Figs. 1E and 2C), and a CD4$^+$ T cell response to OC43 that cross-reacts with SARS-CoV-2 (Figs. 4 and S3G). To determine whether OC43 pre-exposure can protect against SARS-CoV-2, HLA-DRB1\*0101 *Ifnar1*$^{-/-}$ mice were infected with OC43, challenged with SARS-CoV-2 16 days later (SARS-CoV-2 cross-reactive Th1 CD4$^+$ T cell response peaked at 16 days post-OC43 infection; Fig. 4C), and lungs were harvested at 3 days post-challenge (Fig. 6A). Prior OC43 exposure led to dramatically lower levels of SARS-CoV-2 infection in the lungs, based on RT-qPCR analysis of SARS-CoV-2 genomic RNA and immunostaining for N protein (Fig. 6B, C), but did not significantly affect SARS-CoV-2 RNA levels in nasal turbinates (Fig. S6A). Histopathological analysis showed that the lungs of OC43-exposed/SARS-CoV-2-challenged mice were much healthier than those of naïve/SARS-CoV-2-challenged mice. Specifically, OC43 pre-exposure led to more bronchioles with clear lumina, viable epithelial cells, and significant improvement in all 5 features (Fig. 6D). Thus, OC43 pre-exposure for 16 days was sufficient to elicit a cross-protective response against SARS-CoV-2 infection and lung disease in HLA-DRB1\*0101 *Ifnar1*$^{-/-}$ mice.

The majority of SARS-CoV-2-specific T cell responses in humans are CD4$^+$ T cells[31,103,104]. To determine whether the protective effects of OC43 pre-exposure were mediated by CD4$^+$ T cells, we repeated these experiments in mice treated with a CD4 T cell-depleting antibody (vs isotype) immediately prior to SARS-CoV-2 challenge (Fig. 6E). Efficient CD4$^+$ T cell depletion, confirmed by flow cytometry (Figure S6C),

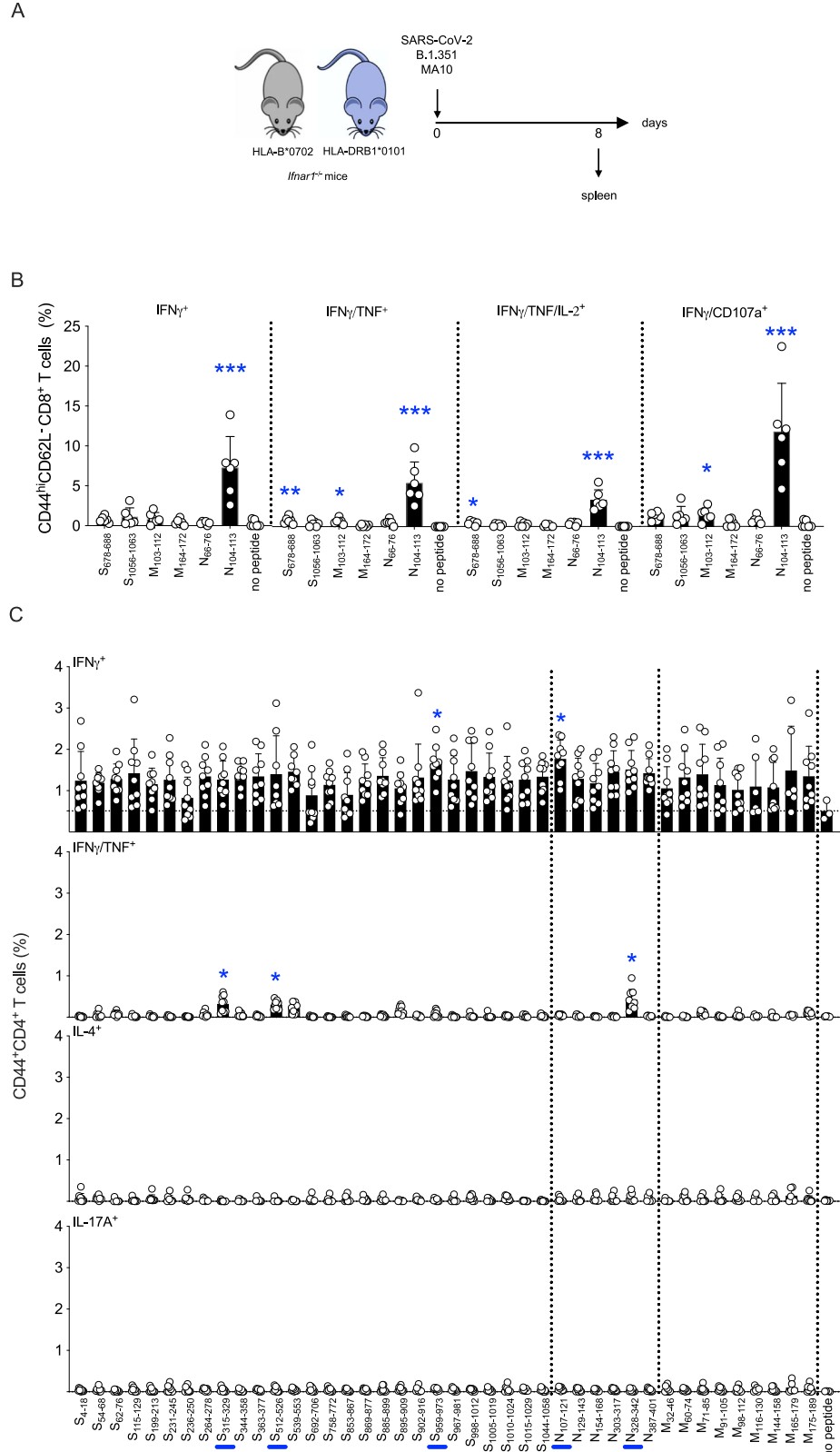

**Fig. 2 | Mapping of SARS-CoV-2 S, N, and M protein-derived epitopes in SARS-CoV-2-infected HLA-B*0702 and HLA-DRB1*0101 *Ifnar1*−/− mice. A** Experimental protocol. HLA-B*0702 and HLA-DRB1*0101 *Ifnar1*−/− mice were infected with SARS-CoV-2 strains B.1.351 or MA10, respectively (10⁴ PFU, IN), and spleens collected 8 days later. **B**, **C** ICS analysis of activated CD8⁺ T cells from B.1.351-infected HLA-B*0702 *Ifnar1*−/− mice (**B**) and activated CD4⁺ T cells from MA10-infected HLA-DRB1*0101 *Ifnar1*−/− mice (**C**). Splenocytes were stimulated for 6 h with the indicated 6 (**B**) or 42 (**C**) SARS-CoV-2 peptides (*vs* no peptide), immunolabeled for cell surface markers, intracellular cytokines, and the degranulation marker CD107a, and analyzed by flow cytometry. Data are presented as the mean ± SEM. N values: 6 (**B**) and 9 (**C**) mice/group, pooled from 2 independent experiments. Peptide vs control were compared using the nonparametric Kruskal–Wallis test. *P < 0.05; **P < 0.01; ***P < 0.001. Circles, individual mice. Blue bars on the *x*-axis, peptides that significantly stimulated CD4⁺ T cells with at least 1 secretion phenotype.

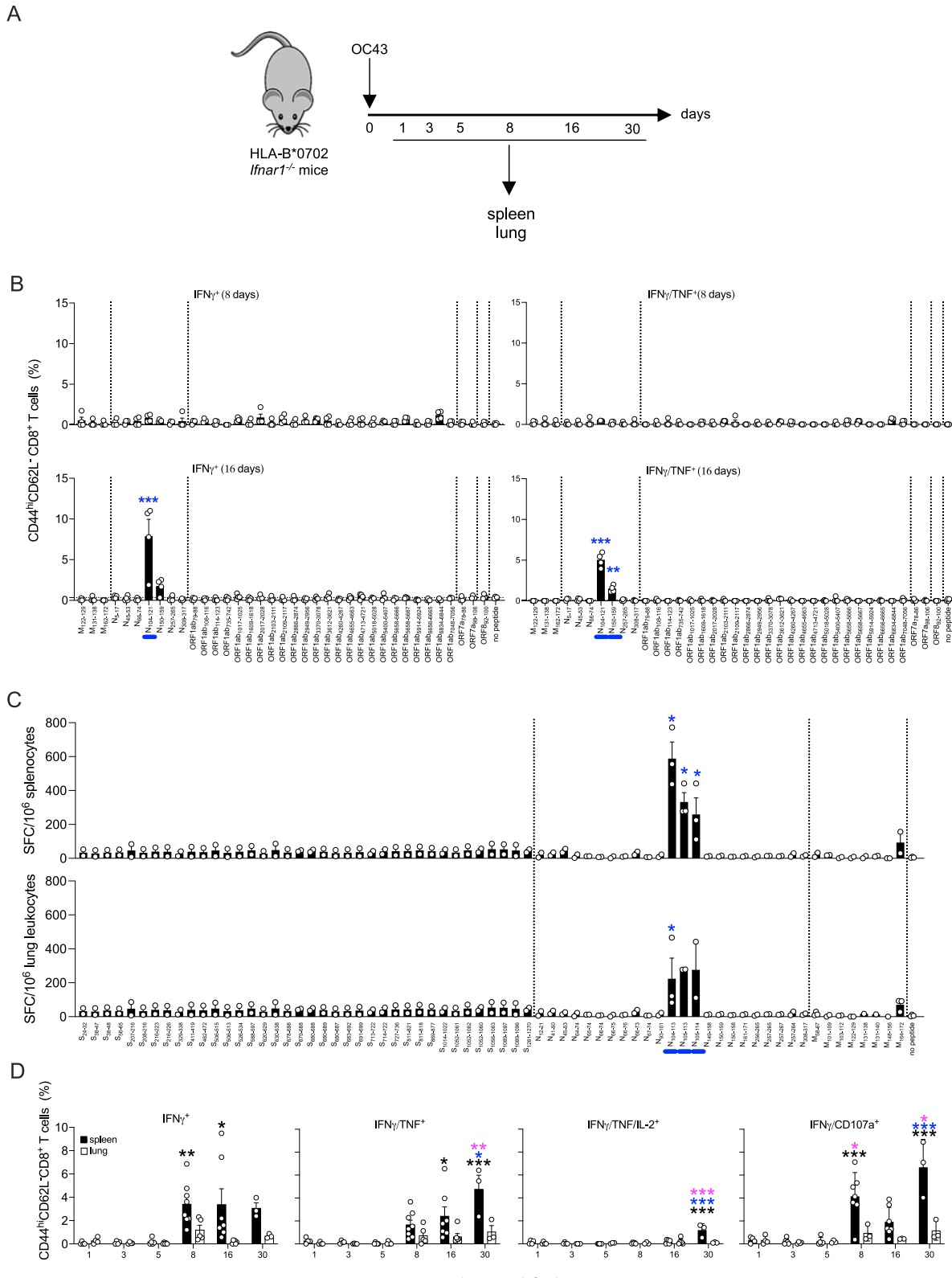

significantly reduced the protective effect of prior OC43 exposure on SARS-CoV-2 genomic RNA levels and N protein expression in lungs (Fig. 6F, G). However, CD4+ T cell-depleted, and isotype control mice (both OC43 infected) showed indistinguishable features of mild pneumonia in the lungs (Fig. 6H) and no difference in SARS-CoV-2 genomic RNA levels in nasal turbinates (Figure S6B). This indicates that OC43-elicited CD4+ T cells contribute to cross-protection against SARS-CoV-2 infection but do not significantly affect lung disease at day 3 after infection.

## Discussion

The emergence of novel SARS-CoV-2 variants, increasingly prevalent infections and breakthrough infections have highlighted the relatively narrow immune response elicited by the currently available SARS-CoV-

**Fig. 3 | Cross-reactivity of OC43-elicited CD8⁺ T cells for SARS-CoV-2 peptides.**
**A** Experimental protocol. HLA-B*0702 *Ifnar1⁻/⁻* mice were infected with OC43 (10⁹ genomic equivalents (GE), IN), and tissues were collected on multiple days. **B** ICS analysis of activated CD8⁺ T cells in splenocytes stimulated for 6 h with 1 of 37 published HLA-B*0702-restricted SARS-CoV-2-derived peptides (Table 2) *vs* no peptide, immunolabeled for cell surface markers and intracellular cytokines, and analyzed by flow cytometry. Circles, individual mice; $N = 4$ mice/group. **C** ELISpot quantification of IFNγ-producing SFCs in splenocytes and lung leukocytes isolated on day 8 post-infection and stimulated for 20 h with 69 SARS-CoV-2 peptides (Table 1) *vs* no peptide. $N = 6$ mice/group. **D** ICS analysis of activated CD8⁺ T cells in splenocytes (black bars) and lung leukocytes (white bars) isolated on multiple days.

Cells were stimulated for 6 h with SARS-CoV-2 $N_{104-113}$ peptide, immunolabeled for cell surface markers, intracellular cytokines, and CD107a, and analyzed by flow cytometry. Circles, individual mice; $N = 4, 4, 5, 8, 7$, and 3 mice/group for spleen samples and $N = 4, 4, 5, 5, 6$, and 3 for lung leukocytes samples from days 1, 3, 5, 8, 16, and 30, respectively. Black, blue, and pink asterisks, comparisons *vs* day 1, day 8, and day 16 data, respectively. **B**–**D** Data, pooled from two independent experiments, are presented as the mean ± SEM, and compared by either the nonparametric Kruskal–Wallis test (**B**, **C**) or two-way ANOVA with Sidak's multiple comparison test (**D**). \*$P < 0.05$; \*\*$P < 0.01$; \*\*\*$P < 0.001$. Blue bars on the *x*-axis in **B** and **C**, peptides that significantly stimulated 1 or more cell groups.

2 vaccines. A substantial fraction of these breakthrough infections occurred in immunocompromised people, in which viral evolution may lead to the generation of new VOCs[105]. In contrast to the anti-SARS-CoV-2 antibody response, which is focused on the highly variable S protein, T cells contribute to protection against SARS-CoV-2 by

### Table 3 | HLA-B*0702-restricted CD8⁺ T cell epitopes identified in the literature (May 2021)

| Protein | Start | End | Length | Sequence |
|---|---|---|---|---|
| Membrane | 122 | 129 | 8 | VPLHGTIL |
| | 131 | 138 | 8 | RPLLESEL |
| | 164 | 172 | 9 | LPKEITVAT |
| Nucleocapsid | 5 | 17 | 13 | GPQNQRNAPRITF |
| | 45 | 53 | 9 | LPNNTASWF |
| | 66 | 74 | 9 | FPRGQGVPI |
| | 104 | 121 | 18 | LSPRWYFYYLGTGPEAGL |
| | 150 | 159 | 10 | NPANNAAIVL |
| | 257 | 265 | 9 | KPRQKRTAT |
| | 308 | 317 | 10 | APSASAFFGM |
| ORF1ab | 79 | 88 | 10 | APHGHVMVEL |
| | 108 | 116 | 9 | VPHVGEIPV |
| | 114 | 123 | 10 | IPVAYRKVLL |
| | 735 | 742 | 8 | APKEIIFL |
| | 1017 | 1025 | 9 | TPVVQTIEV |
| | 1608 | 1618 | 11 | KPHNSHEGKTF |
| | 2017 | 2028 | 12 | KPVETSNSFDVL |
| | 2109 | 2117 | 9 | KPNELSRVL |
| | 2703 | 2711 | 9 | VAKSHNIAL |
| | 2866 | 2874 | 9 | VPGLPGTIL |
| | 2949 | 2956 | 8 | RPDTRYVL |
| | 3370 | 3378 | 9 | QPGQTFSVL |
| | 3612 | 3621 | 10 | LPFAMGIIAM |
| | 4260 | 4267 | 8 | VPANSTVL |
| | 4655 | 4663 | 9 | KPYIKWDLL |
| | 4713 | 4721 | 9 | FPPTSFGPL |
| | 5018 | 5028 | 11 | MPNMLRIMASL |
| | 5400 | 5407 | 8 | KPPISFPL |
| | 5658 | 5666 | 9 | IPARARVEC |
| | 5658 | 5667 | 10 | IPARARVECF |
| | 5914 | 5924 | 11 | LEIPRRNVATL |
| | 6656 | 6665 | 10 | KPRSQMEIDF |
| | 6834 | 6844 | 11 | LPKGIMMNV |
| | 7048 | 7056 | 9 | FPLKLRGTA |
| ORF7 | 78 | 86 | 9 | RARSVSPKL |
| | 98 | 106 | 9 | SPIFLIVAA |
| ORF8 | 92 | 100 | 9 | EPKLGSLVV |

recognizing conserved epitopes of multiple SARS-CoV-2 proteins[106–110], particularly in the context of impaired humoral immunity[111–116]. In addition, T cells that recognize homologous epitopes between seasonal HCoVs and SARS-CoV-2 are present in some healthy individuals previously unexposed to SARS-CoV-2[18,21,23,29,82,93,117,118]. Importantly, robust pre-existing HCoV/SARS-CoV-2 cross-reactive T cell responses that are rapidly induced following SARS-CoV-2 exposure have been associated with asymptomatic infection and less severe COVID-19, suggesting a role for these cells in protective immunity to SARS-CoV-2[23,29,39,41,81,119]. Therefore, understanding the pre-existing SARS-CoV-2 cross-reactive T cell responses in SARS-CoV-2 naïve individuals helps explain the broad heterogeneity in COVID-19 outcomes and to develop pan-CoV vaccines that could provide broad protection against current and future SARS-CoV-2 variants and HCoVs.

Multiple human studies suggest that pre-existing SARS-CoV-2 cross-reactive T cells derive, partly or wholly, from exposure to seasonal HCoVs[39,120,121] and can have either protective or pathogenic consequences for subsequent SARS-CoV-2 infection. To test this directly, we used HLA-B*0702 and HLA-DRB1*0101 transgenic *Ifnar1⁻/⁻* mice. We found that primary SARS-CoV-2 infection or vaccination with SARS-CoV-2 protein-encoding vectors elicited CD8⁺ and CD4⁺ T cell responses that recapitulated the epitope specificity, Tc1- and Th1-bias, and mono-functional/multifunctional phenotypes observed in SARS-CoV-2-infected or vaccinated humans[18,21,23,29,82,93,117,122]. OC43-SARS-CoV-2 sequential infection of our mice showed that OC43 pre-exposure was protective against SARS-CoV-2, and this protection was partly dependent on OC43-elicited CD4⁺ T cells. It is plausible that additional elements of the pre-existing OC43 immunity, such as CD8⁺ T cells, may also contribute to this protective response. Our findings present direct experimental evidence for a protective role for seasonal HCoV/SARS-CoV-2 cross-reactive CD4⁺ T cell responses, consistent with human data implicating cross-reactive CD4⁺ T cells in protecting against SARS-CoV-2 infection[23,29,39,40,42,119,123,124]. They also align with a mouse study demonstrating that CD4⁺ T cells recognizing a conserved epitope present in SARS-CoV, MERS-CoV, and bat CoV HKU4 are cross-protective[125]. We detected minimal cross-reactive humoral response to SARS-CoV-2 in HLA-DRB1*0101 transgenic *Ifnar1⁻/⁻* mice pre-exposed to OC43, which is consistent with data showing no significant cross-reactivity of human anti-HCoV antibodies with SARS-CoV-2[126,127], and broadly coronavirus-reactive antibodies in very few SARS-CoV-2-immune individuals[128–131]. In fact, human studies have reported that the weak, cross-reactive antibody response decays rapidly in most individuals, and the cross-reactive cellular but not humoral immunity likely contributes to significant protection against SARS-CoV-2[18,132–135]. In line with the report that the CD4⁺ T cell cross-reactivity to SARS-CoV-2 in healthy SARS-CoV-2-unexposed individuals is variable and low, and that the degree of amino acid sequence identity between common cold HCoVs and SARS-CoV-2 does not correlate with CD4⁺ T cell cross-reactivity[36], some of the OC43-elicited SARS-CoV-2 cross-reactive CD4⁺ T cell specificities in our mouse model are of low magnitude and share limited amino acid sequence homology between OC43 and SARS-CoV-2.

The level of protection mediated by pre-existing, cross-reactive T cells may be influenced by modulation of immunodominance

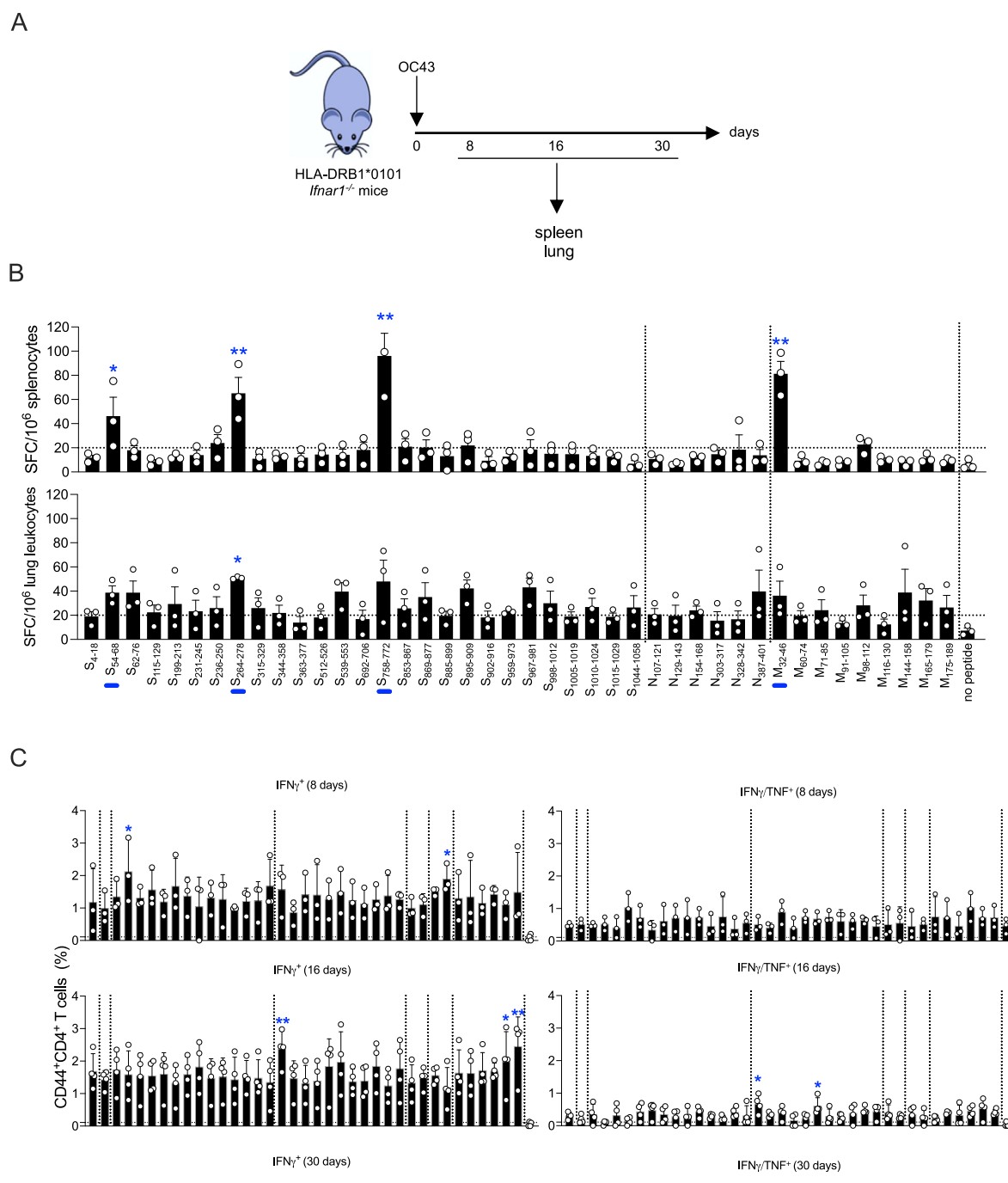

**Fig. 4 | Cross-reactivity of OC43-elicited CD4⁺ T cells for SARS-CoV-2 peptides.**
**A** Experimental protocol. HLA-DRB1*0101 *Ifnar1⁻/⁻* mice were infected with OC43 (10⁹ GE, IN), and tissues were collected 8, 16, and 30 days later. **B** ELISpot quantification of IFNγ-producing SFCs in splenocytes and lung leukocytes isolated on day 8 post-infection and stimulated for 20 h with 42 SARS-CoV-2 peptides (Table 2) *vs* no peptide. *N* = 8 mice/group. **C** ICS analysis of activated CD4⁺ T cells in splenocytes isolated on days 8, 16, and 30 post-infection. Cells were stimulated for 6 h with the

37 published HLA-DRB1*0101-restricted SARS-CoV-2-derived peptides (Table 4) *vs* no peptide, immunolabeled for cell surface markers and intracellular cytokines, and analyzed by flow cytometry. Circles, individual mice; *N* = 4 mice/group.
**B**, **C** Data, pooled from two independent experiments and presented as the mean ± SEM, were compared by the nonparametric Kruskal–Wallis test. *\*P* < 0.05; *\*\*P* < 0.01. Blue bars on the *x*-axis are peptides that significantly stimulated 1 or more cell groups.

**Table 4 | HLA-DRB1*0101-restricted CD4+ T cell epitopes identified in the literature (May 2021)**

| Protein | Start | End | Length | Sequence |
|---|---|---|---|---|
| Spike | 530 | 544 | 15 | STDLIKNQCVNFNFN |
| Envelope | 26 | 40 | 15 | FLLVTLAILTALRLC |
| Membrane | 36 | 50 | 15 | QFAYANRNRFLYIIK |
| | 66 | 80 | 15 | VLAAVYRINWITGGI |
| | 71 | 85 | 15 | YRINWITGGIAIAMA |
| | 86 | 100 | 15 | CLVGLMWLSYFIASF |
| | 91 | 105 | 15 | MWLSYFIASFRLFAR |
| | 116 | 130 | 15 | TNILLNVPLHGTILT |
| | 136 | 150 | 15 | SELVIGAVILRGHLR |
| | 146 | 160 | 15 | RGHLRIAGHHLGRCD |
| | 151 | 165 | 15 | IAGHHLGRCDIKDLP |
| | 161 | 175 | 15 | IKDLPKEITVATSRT |
| | 166 | 180 | 15 | KEITVATSRTLSYYK |
| | 176 | 190 | 15 | LSYYKLGASQRVAGD |
| | 191 | 205 | 15 | SGFAAYSRYRIGNYK |
| Nucleoprotein | 81 | 95 | 15 | DDQIGYYRRATRRIR |
| | 86 | 100 | 15 | YYRRATRRIRGGDGK |
| | 126 | 140 | 15 | NKDGIIWVATEGALN |
| | 211 | 225 | 15 | AGNGGDAALALLLLD |
| | 216 | 230 | 15 | DAALALLLLDRLNQL |
| | 221 | 235 | 15 | LLLLDRLNQLESKMS |
| | 261 | 275 | 15 | KRTATKAYNVTQAFG |
| | 301 | 315 | 15 | WPQIAQFAPSASAFF |
| | 317 | 331 | 15 | MSRIGMEVTPSGTWL |
| | 326 | 340 | 15 | PSGTWLTYTGAIKLD |
| | 346 | 360 | 15 | FKDQVILLNKHIDAY |
| | 351 | 365 | 15 | ILLNKHIDAYKTFPP |
| ORF1ab | 5041 | 5055 | 15 | SHRFYRLANECAQVL |
| | 5246 | 5260 | 15 | LMIERFVSLAIDAYP |
| ORF3a | 106 | 120 | 15 | LYLYALVYFLQSINF |
| | 116 | 130 | 15 | QSINFVRIIMRLWLC |
| ORF8 | 36 | 50 | 15 | PCPIHFYSKWYIRVG |
| | 41 | 55 | 15 | FYSKWYIRVGARKSA |
| | 76 | 90 | 15 | IGNYTVSCLPFTINC |
| | 86 | 100 | 15 | FTINCQEPKLGSLVV |
| | 96 | 110 | 15 | GSLVVRCSFYEDFLE |
| | 101 | 115 | 15 | RCSFYEDFLEYHDVR |

hierarchies, as observed during Zika virus infection in humans and HLA transgenic mice with prior exposure to dengue virus[74,136]. Accumulating evidence suggests that immunodominance may be an important feature of the T cell response to SARS-CoV-2. For example, in a single cohort of SARS-CoV-2-infected individuals, researchers identified 2 populations of SARS-CoV-2 S protein-specific CD4+ T cells that were differentially activated: $S_{751}$- and $S_{236}$-specific, which were dominant and subdominant, respectively[137]. In addition, T cells recognizing an immunodominant seasonal HCoV/SARS-CoV-2 cross-reactive epitope restricted by HLA-B*1501 are observed in SARS-CoV-2 naïve and exposed individuals[39]. Similarly, CD8+ T cells recognizing the HLA-B*0702-restricted SARS-CoV-2 epitope $N_{105-113}$ (the most immunodominant SARS-CoV-2 CD8+ T cell epitope identified to date) are present at high frequencies in unexposed healthy individuals[25,28,79–82]. Consistent with these human data, we found that CD8+ T cells specific for or cross-reactive with SARS-CoV-2 $N_{104/105-113}$ epitope were immunodominant. Importantly, we observed that immunization of naïve HLA-B*0702 transgenic *Ifnar1*−/− mice with the single SARS-CoV-2 $N_{104-113}$

peptide evoked a response that limited SARS-CoV-2-induced pathogenesis in the lung. Thus, the presence of immunodominant T cells within pre-existing SARS-CoV-2 cross-reactive T cell immunity could help to explain the disparate outcomes of COVID-19 patients.

A study in humans showed that HLA genotype shapes SARS-CoV-2-specific and memory cross-reactive CD8+ T cell responses[138], and another recent study reported that HLA-B*1501 associates with asymptomatic SARS-CoV-2 infection in humans[39]. Our findings in mice expressing a single HLA allele provide a validated tool for investigating how allelic variation in HLA and infection parameters (i.e., viral strains, number and sequence of infections, and the interval between infections) influence cross-reactive T cell responses during sequential infections with seasonal HCoV and SARS-CoV-2. Our data show that a single pre-exposure to OC43 can confer cross-protection against SARS-CoV-2. However, depending on the HLA alleles and infection parameters, pre-existing HCoV-elicited T-cell immunity may also play a pathogenic role. This is supported by studies suggesting that cross-reactive CD8+ T cells (vs SARS-CoV-2-specific) have lower affinity[91], and low-avidity SARS-CoV-2 cross-reactive CD4+ T cells have a reduced ability to proliferate in response to SARS-CoV-2 and are associated with more severe COVID-19[19,38]. The present study sets a framework for investigating the role of HCoV-cross-reactive T cell responses using various HLA-expressing mice, HCoVs, and infection parameters, including sequential infections with SARS-CoV-2 followed by OC43.

In summary, the current study demonstrates a protective role for pre-existing seasonal HCoV/SARS-CoV-2-cross-reactive CD4+ T cell responses during SARS-CoV-2 infection. As human studies suggest both a protective and pathogenic role for HCoV-cross-reactive CD4+ T cell responses, a greater understanding of these T cell responses is required to facilitate T cell epitope-based rational vaccine design[139]. SARS-CoV-2 vaccines that are effective in individuals with antibody or B cell deficiencies are urgently needed to protect this highly vulnerable population and limit the emergence of new VOCs. Similarly, pan-HCoV vaccines that elicit cross-protective T cell responses represent key tools against new SARS-CoV-2 VOCs and HCoVs with pandemic potential.

**Limitations.** Our study has several limitations. First, our findings are based on two HLA transgenic mouse strains that express a single HLA class I or II molecule, but not both, and lack a human TCR repertoire. Therefore, our results may not reflect the full spectrum of human T-cell responses. Second, OC43 RNA was detected only transiently in the lungs following infection of our two mouse strains; therefore, our observations reflect the immune response following spatially and temporally restricted exposure to OC43 antigens. Third, along similar lines, we examined the animals at 3 days post-infection in order to observe the cross-reactive memory response; however, this time point may have been suboptimal for assessing changes in lung pathology. Fourth, although type I IFN receptor deficiency has been observed in nearly 20% of severe COVID patients[140], anti-OC43, and anti-SARS-CoV-2 T cell responses may differ quantitatively and/or qualitatively in wildtype vs *Ifnar1*−/− mice. Finally, we focused our study solely on OC43 seasonal HCoV infection prior to SARS-CoV-2 challenge, which does not model epidemiologic scenarios in which human adults are exposed to multiple seasonal HCoVs prior to SARS-CoV-2 infection. Despite these limitations, a major strength of our study is the use of mouse models in which infection parameters can be precisely controlled. Our study establishes a foundation for conducting studies using mouse models of sequential infections with NL63, 229E, or HKU1 followed by SARS-CoV-2 and investigating specific conditions and mechanisms by which HCoV-cross-reactive T cell responses contribute to protective or pathogenic immunity. Next-generation mice that co-express HLA molecules and human T cell antigen receptors[141] are poised to further advance our understanding of the factors that dictate the heterogeneity of COVID-19 clinical and immunological outcomes.

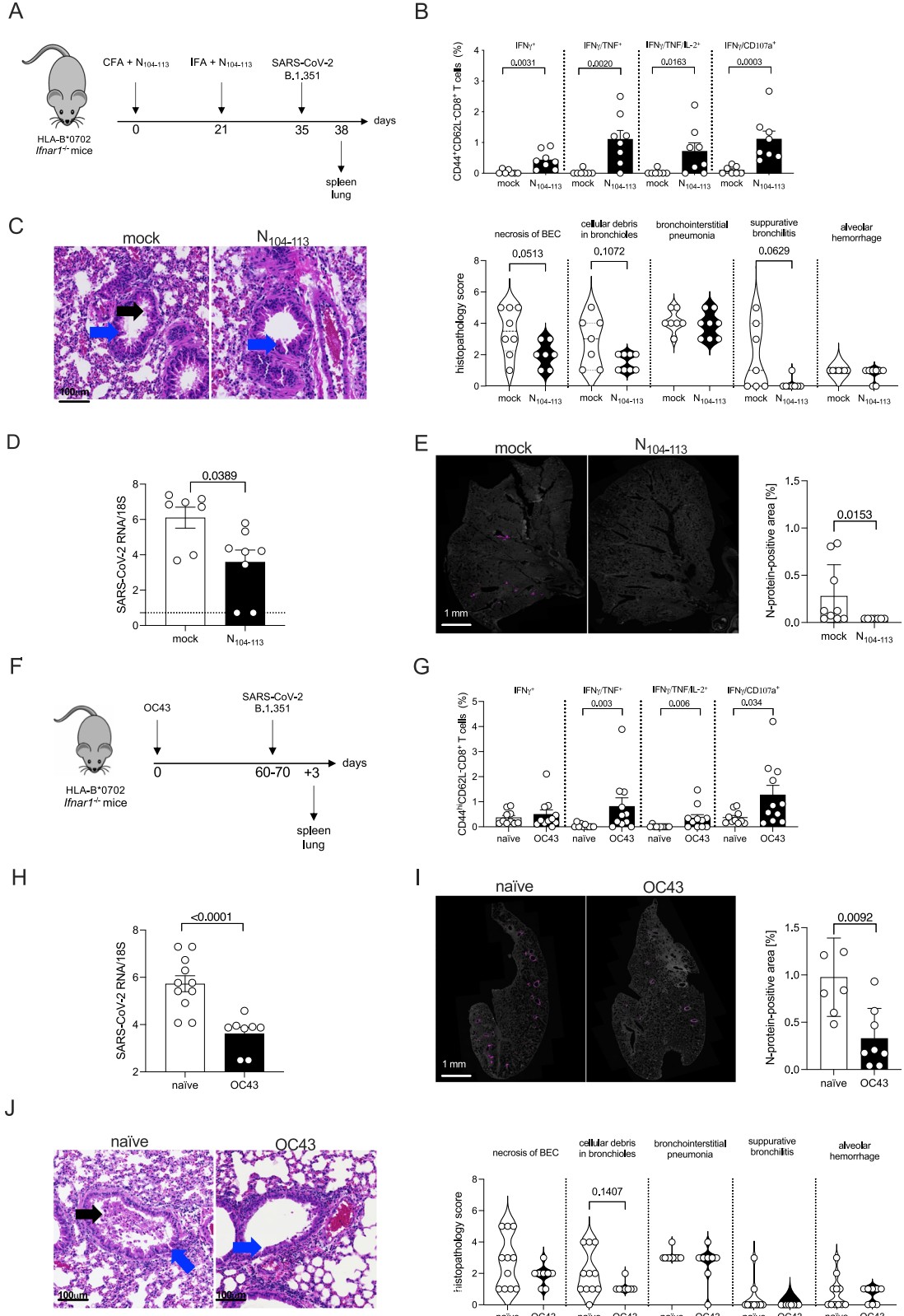

## Methods

### Study design

Numerous human cohort studies have revealed that SARS-CoV-2-unexposed individuals harbor CD8[+] and CD4[+] T cells that recognize peptides present in both SARS-CoV-2 and seasonal HCoV. These pre-existing cross-reactive T cells have been associated with both protective and pathologic immunity during SARS-CoV-2 infection, suggesting that they may play an important role in dictating infection outcome, which can range from asymptomatic or mild symptoms to severe COVID-19 and death. The objective of this study was to dissect the origin of pre-existing T cells that cross-react with SARS-CoV-2 and their roles during SARS-CoV-2 infection, in terms of viral load and disease outcomes.

**Fig. 5 | Protective effect of OC43 pre-exposure and SARS-CoV-2 N₁₀₄₋₁₁₃ immunization on SARS-CoV-2 infection and lung disease in HLA-B*0702 *Ifnar1*⁻/⁻ mice. A** Experimental protocol for B to E. Mice were injected with SARS-CoV-2 N$_{104-113}$ *vs* DMSO (mock) on day 0 (complete Freund's adjuvant, CFA) and again on day 21 (incomplete Freund's adjuvant, IFA). Two weeks later, mice were challenged with SARS-CoV-2 B.1.351 (10⁵ PFU, IN), and tissues were collected at 3 days post-challenge. Mice/group: 8 (peptide-immunized) and 7 (mock). **B** ICS analysis of activated CD8⁺ T cells. Splenocytes were stimulated for 6 h with SARS-CoV-2 N$_{104-113}$ peptide, immunolabeled for cell surface markers, intracellular cytokines, and CD107a, and analyzed by flow cytometry. $N = 7$ and $N = 8$, respectively, for the mock and N$_{104-113}$ groups. **C** Representative H&E-stained sections of lungs. Blue arrows, bronchiolar epithelial cells (BEC); black arrows, epithelial cells within bronchioles. Sections were scored from 0 (least severe) to 5 (most severe) for standard histopathological features of SARS-CoV-2-induced lung disease. **D, E** RT-qPCR of SARS-CoV-2 genomic RNA in lungs, and representative immunofluorescence of SARS-CoV-2 N protein (magenta) in lung sections with quantification of the N protein staining. In figure D, $N = 7$ and $N = 8$, respectively, for the mock and N$_{104-113}$ groups. In figure E, $N = 9$ and $N = 7$, respectively, for the mock and N$_{104-113}$ groups. **F** Experimental protocol for G to J. Mice were infected with OC43 (10⁹ GE, IN) *vs* PBS (naïve) and challenged with SARS-CoV-2 B.1.351 (10⁵ PFU, IN) 60–70 days later. Tissues were collected 3 days post-challenge. **G** ICS analysis of activated CD8⁺ T cells as described in B. Mice/group: 11 (OC43-infected) and 10 (naïve). **H, I** RT-qPCR of genomic SARS-CoV-2 RNA in lungs, and N protein staining in lung as described in D and E. Mice/group: 8 (OC43-infected) and 7-11 (naïve). **J** Lung histopathology and scoring as described in C. Mice/group: 8 (OC43-infected) and 11 (naïve). **B, D, E, G–I** Data pooled from 2 to 3 independent experiments and presented as the mean ± SEM. **C, J** Data pooled from 2 independent experiments and presented as violin plots. The means were compared using the two-sided Mann–Whitney test. Exact *P* values are indicated directly on the figure. Circles, individual mice.

## Mice

For this study, we developed a mouse model in which HLA class I (HLA-B*0702 *Ifnar1*⁻/⁻) or class II (HLA-DRB1*0101 *Ifnar1*⁻/⁻) transgenic mice are infected sequentially with 1 of the 4 major seasonal human coronaviruses followed by SARS-CoV-2. Both transgenic mice were bred under pathogen-free conditions at La Jolla Institute for Immunology. The sex ratio for all experiments was approximately 1:1, and all experiments were started when mice were 5 to 7 weeks of age. For tissue collection, mice were euthanized by $CO_2$ inhalation. Blood samples were collected into serum collection tubes (Sarstedt) from a facial vein/cardiac puncture in ABSL2 or terminal eye bleeding in ABSL3. All samples were included in the analyses unless technical issues were evident, such as >50% cell mortality after cell isolation.

SARS-CoV-2 infections were performed in our high containment facility and OC43 infections in our biosafety level 2 infectious facility. All experiments were performed in strict accordance with recommendations set forth in the National Institutes of Health Guide for the Care and Use of Laboratory Animals and approved by the Institutional Animal Care and Use Committee at the La Jolla Institute for Immunology ABSL2 and ABSL3 (protocol number AP00001242). The study design was approved by the respective affiliated institutions of all authors.

## Vaccination and infection

Mice were vaccinated IM (quadriceps) via electroporation with a minimally invasive device[142] (BTX Agile Pulse system [47-0500 N] with a 4 × 4 × 5 mm needle array [47-0045]) with 25 µg of S, M, or N DNA vaccine and boosted 14 days later in the same manner. For vaccination with SARS-CoV-2 N$_{104-113}$ peptide, the peptide (250 µg) was diluted in PBS, homogenized in complete Freund's adjuvant (CFA), and injected IM (the site was gently massaged to facilitate dispersion of the peptide vaccine), and boosted 3 weeks later in the same manner homogenized in incomplete Freund's adjuvant (IFA). For the mock-vaccinated mice, peptide was replaced with DMSO.

Mice were infected IN with 10⁹ GE of OC43 (ATCC, VR-1558), 10⁴ PFU of SARS-CoV-2 MA10 (Leist et al., 2020), or 10⁵ PFU of SARS-CoV-2 B.1.351 (isolate HCoV-19/South Africa/KRISP-K005325/2020, NR-54009); both SARS-CoV-2 strains were obtained from BEI Resources (NIAID, NIH).

## DNA vaccine constructs and detection of viral proteins

Plasmids encoding SARS-CoV-2 S, M, or N proteins (SARS-CoV-2/human/USA/WA-CDC-WA1/2020 isolate, GenBank MN985325.1) were synthesized using human codon optimization. Optimized DNA sequences were synthesized (GenScript), digested with KpnI and NotI, and cloned into pVAX1 under the control of human cytomegalovirus immediate-early promoter with a bovine growth hormone polyadenylation signal and kanamycin as a resistance marker. To increase efficiency of translational initiation, Kozak and IgE leader sequences were introduced. An empty pVAX1 vector served as a negative control.

For transfection, 293 T cells were seeded at $2 \times 10^5$ cells/well in 24-well plates in DMEM supplemented with 10% fetal bovine serum (FBS), 1% penicillin–streptomycin, and 1% HEPES buffer, and grown to 70%-80% confluence (37 °C, 5% $CO_2$). One hour before transfection, the supernatant was removed, 200 µl of Opti-MEM™ (Thermo Fisher Scientific) was added, and cell monolayers transfected with DNA vaccines or empty pVAX1 mixed with Lipofectamine® 2000 (1.5 µg/3 µL) per manufacturer's instructions. After 20 hours, cells were rinsed 3 times with PBS, and protein expression was assessed by immunofluorescence microscopy or flow cytometry.

For microscopy, monolayers were fixed with Cytofix (BD Biosciences) per manufacturer's instructions, permeabilized with 0.1% Triton X-100 (for S and N proteins) or 0.05% saponin (for M protein) in PBS (10 min, room temperature), blocked with 3% BSA in PBS (30 min, room temperature), and immunolabeled with mouse monoclonal antibodies (1 h, room temperature) against S protein (Thermo Fisher Scientific, MA5-35946, RRID AB2866558) or N protein (Absolute Antibody, Ab01691-3), or a polyclonal antibody against M protein (Prosci, 9157) diluted 1:500 in 0.1% Triton X-100/PBS. Monolayers were then washed 3 times with PBS, incubated (1 h, room temperature) with Alexa Fluor 488-conjugated goat anti-mouse IgG (Thermo Fisher Scientific, A11001) diluted 1:200 in 3% BSA/PBS, washed 3 times with PBS, and overlaid with a drop of ProLong™ Gold Antifade Mountant (Thermo Fisher Scientific). Images were captured with a Keyence BZ-X810 fluorescence microscope using with a Plan Fluor 20X/0.5 dry objective.

For flow cytometry, cell monolayers were trypsinized, fixed, permeabilized with Cytofix/Cytoperm (BD Biosciences), and incubated with the anti-N protein and secondary antibodies as described above (30 min, 4 °C). Cells were washed twice with Cytoperm containing 0.1% BSA and once with FACS buffer, and then resuspended in FACS buffer. Data were collected on an LSR II flow cytometer (BD Biosciences) and analyzed using FlowJo software.

## Virus propagation and titration

OC43 was propagated for 9 days in HCT-8 cells cultured in RPMI supplemented with 10% FBS, 1% penicillin–streptomycin, and 1% HEPES buffer. The supernatant was collected and virus was concentrated using a gradient-free method with an Amicon Ultra-15 centrifugal filter unit (Millipore Sigma, UFC9100). Each virus batch was titrated, as described previously[143], by amplifying the M protein gene using genomic RT-qPCR and the following primers: Rev, 5′-AAT GTA AAG ATG GCC GCG TAT T-3′; Fwd, 5′-ATG TTA ACC TT TAA TTG AGG ACT AT-3′ (IDT Integrated DNA Technologies). Cycling conditions were as follows: transcription initiation (48 °C, 30 min), PCR activation (95 °C, 10 min), and 45 cycles of amplification (95 °C for 15 s and 60 °C for 1 min). Viral RNA concentration was calculated using a standard curve composed of at least 4 100-fold serial dilutions of in vitro-transcribed OC43 RNA.

SARS-CoV-2 MA10 and B.1.351 were propagated for 3 days in Vero cells (ATCC, CCL81) cultured in Dulbecco's Modified Eagle's Medium

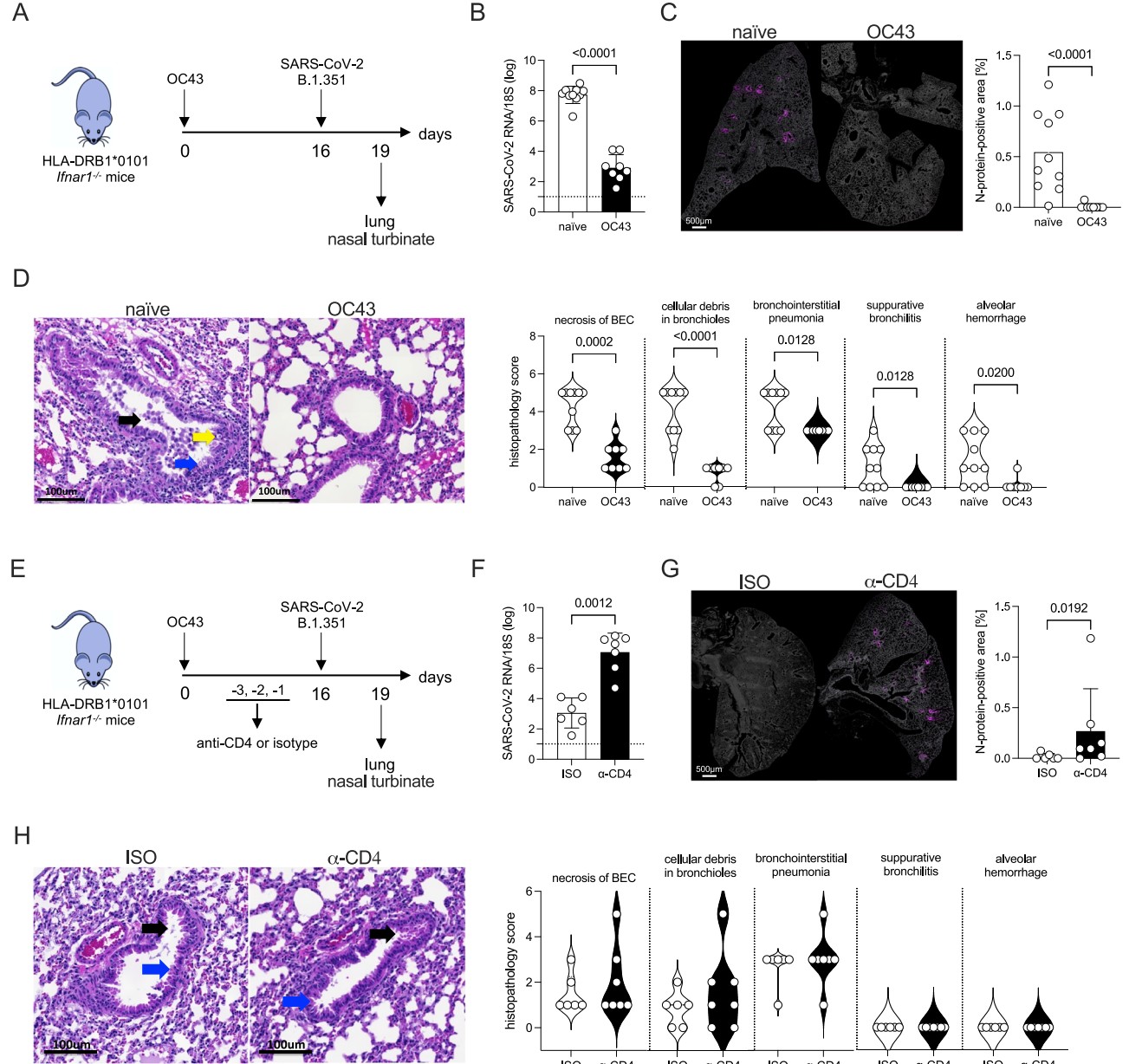

**Fig. 6 | Protective effect of OC43 pre-exposure on SARS-CoV-2 infection and lung disease in HLA-DRB1*0101 *Ifnar1*⁻/⁻ mice. A** Experimental protocol for B to D. Mice were infected with OC43 ($10^9$ GE, IN) *vs* PBS (naïve), challenged with SARS-CoV-2 B.1.351 ($10^5$ PFU) 16 days later, and lungs collected at 3 days post-challenge. **B**, **C** RT-qPCR of SARS-CoV-2 genomic RNA, and representative SARS-CoV-2 N protein (magenta) immunofluorescence in sections with quantification of the N protein staining. $N = 10$ and $N = 8$, respectively, for the naïve and OC43 groups. **D** Representative H&E-stained sections of lungs. Blue arrow, bronchiolar epithelial cells (BEC); black arrow, epithelial cells within bronchioles; yellow arrow, perivascular cuffing. Sections were scored from 0 (least severe) to 5 (most severe) for standard histopathological features of SARS-CoV-2-induced lung disease. $N = 10$ and $N = 8$, respectively, for the naïve and OC43 groups. **E** Experimental protocol for

**F** to **H**. Mice were infected with OC43 ($10^9$ GE, IN) *vs* PBS (naïve) and challenged with B.1.351 ($10^5$ PFU, IN) 16 days later. Mice were injected (IP) with CD4⁺ T cell-depleting antibody (α-CD4) *vs* isotype control antibody (ISO) once daily for 3 days before the challenge. Lungs were collected 3 days post-challenge. **F**, **G** RT-qPCR of genomic SARS-CoV-2 RNA in lungs, and N protein staining in the lung as described in B and C. $N = 6$ and $N = 7$, respectively, for the ISO and α-CD4 groups. **H** Lung histopathology and scoring as described for **D**. $N = 6$ and $N = 7$, respectively, for the ISO and α-CD4 groups. **B–D**, **F–H** Data, pooled from 2–3 independent experiments and presented as the mean ± SEM or violin plots, were compared using the two-sided Mann–Whitney test. Exact *P* values are indicated directly on the figure. Circles, individual mice.

(Corning) supplemented with 10% FBS, 1% penicillin–streptomycin, 1% HEPES buffer, and 1% non-essential amino acids. The supernatant was harvested and titrated using a plaque assay[144]. Briefly, 10-fold serially diluted viral supernatants were added to confluent Vero E6 cells ($8×10^4$ cells/well in 24-well plates; 2 h at 37 °C), supernatants removed, 1% carboxymethylcellulose medium added, and the cells incubated for 3 days. The cells were then fixed with 10% formaldehyde (1 h, room temperature) and stained with 0.1% crystal violet (20 min, room temperature). Viral stocks were deep-sequenced by the La Jolla Institute for Immunology Sequencing Core.

## Quantification of viral RNA in tissues

Organs were harvested, placed in 1 mL RNA/DNA shield (ZYMO Research, R1100-250) for 2 h to maintain high-quality RNA and inactivate the virus, and then transferred into RLT lysis buffer containing 1% 2-mercaptoethanol and homogenized (30 Hz, 3 min) using a Tissue

Lyser II (QIAGEN). Total RNA was extracted using a RNeasy Mini Kit (QIAGEN) and stored at −80 °C. SARS-CoV-2 genomic E RNA and sub-genomic 7a RNA were quantified by RT-qPCR using the qScript One-Step qRT-PCR Kit (Quantabio). For the E gene, the following published primer sets[145] were used: Fwd, 5′-ACA GGT ACG TTA ATA GTT AAT AGC GT-3′; Rev, 5′-ATA TTG CAG TAC GCA CAC A-3′; and Probe, FAM-ACA CTA GCC ATC CTT ACT GCG CTT CG-BBQ. For the 7a gene, modified primer sets[146] were used: Fwd, 5′-TCC CAG GTA ACA AAC CAA CCA ACT-3′; Rev, 5′-AAA TGG TGA ATT GCC CTC GT-3′; and Probe, FAM-CAG TAC TTT TAA AAG ACC TT GCT CTT CTG GAA C-Tamra-Q. Viral RNA concentration was calculated using a standard curve derived from 4 10-fold serial dilutions of in vitro-transcribed SARS-CoV-2 RNA (from isolate USA-WA1/2020, ATCC NR-52347).

## Production of recombinant SARS-CoV-2 S and N proteins

For SARS-CoV-2 S protein, HEK-293F cells were cultured to approximately $3×10^6$ cells/mL, transfected with 3 µg/mL of Hexapro-Spike DNA; the S DNA was first mixed with 9 µg/mL of PEI-MAX (Polysciences), and shaken for 4–5 days (37 °C, 80% humidity, 5% $CO_2$). When cell viability had decreased to 80%, the supernatant was harvested, centrifuged (6000×g, 20 min), mixed with Biolock reagent (IBA Lifesciences, 2-0205-050; 1:300 v/v), stirred (15 min to overnight, 4 °C), and centrifuged again (6000×g, 30 min) to remove Biolock-conjugated biotin. From this clarified supernatant, S protein was purified by affinity chromatography using a Strep-Tactin column (IBA Lifesciences) on an AKTA purifier (GE Healthcare). Protein fractions were pooled and concentration estimated by UV absorbance at 280 nm. Tags were removed by addition of HRV-3C protease (10% w/w), and the digested protein further purified by size-exclusion chromatography using tandem Superose S6 Increase columns (GE Healthcare). The purified untagged S protein was concentrated using Vivaspin 500-10 K filters (Sartorius), aliquoted, flash-frozen in liquid nitrogen, and stored at −80 °C.

Codon-optimized SARS-CoV-2 N was cloned into pET46 vector (Novagen) with an upstream hexahistidine tag followed by an enterokinase and tobacco etch virus (TEV) cleavage site. Plasmid (100 ng) was transformed by heat shock in Rosetta2 pLysS E. coli (Novagen), and starter cultures grown with 20 g/mL chloramphenicol and 100 g/mL ampicillin in 50 mL Luria-Bertani broth (LB) (37 °C). After 14-16 hours, these starter cultures were used to inoculate 1 L LB cultures containing 100 g/mL ampicillin. When the optical density at 600 nm ($OD_{600}$) reached ~0.6, protein production was induced by the addition of 0.5 mM isopropyl-D-thiogalactopyranoside followed by incubation (16–18 h, 25 °C). Cells were then pelleted, resuspended in binding buffer (50 mM Tris-HCl, pH 8.0, 300 mM NaCl, and 30 mM imidazole) supplemented with 500 U of benzonase (Biotool, B16012) and protease inhibitors (AEBSF, E64, pepstatin A), and lysed using a Microfluidics M-110P microfluidizer. Cell debris was removed by centrifugation (25,000 ×g, 25 min) followed by supernatant filtration (0.22 µm pore size). His-coupled SARS-CoV-2 N protein was incubated with nickel-nitrilotriacetic acid beads (1 h), eluted in binding buffer containing TEV protease (1 mg/mL, 0.5% wt/wt) to cleave the His-tag, and dialyzed overnight in snakeskin dialysis tubing (3500 kDa pore size) in 50 mM Tris-Cl, pH 8.5, and 300 mM NaCl. The protein was further purified by size-exclusion chromatography using tandem Superose S6 Increase columns, concentrated using Vivaspin 500-10 K filters (Sartorius), aliquoted, flash-frozen in liquid nitrogen, and stored at −80 °C.

## N and S protein-binding IgG ELISAs

High-binding affinity 96-well plates (Costar) were coated overnight with 1 µg/mL of recombinant SARS-CoV-2 S or N protein or OC43 S protein (Sino Biological, 40607-V08B), and then blocked with 5% blotting-grade casein (Bio-Rad). The following steps were performed at room temperature. Mouse serum samples were diluted 3-fold from 1:30 to 1:810 (S protein) or 1:30 to 1:65,610 (N protein) in 1% BSA/PBS, and added to the coated wells (1.5 h). Plates were washed 3 times with PBS containing 0.05% Tween-20 (pH 7.4), incubated with a 1:5000 dilution of horseradish peroxidase-conjugated anti-mouse IgG polyclonal antibody (Jackson ImmunoResearch) in 1% BSA/PBS, and washed again. Color development was initiated by the addition of TMB substrate (Pierce), and the plates were then incubated in the dark (15 min). The reaction was stopped by the addition of 2 N sulfuric acid (Fisher Chemical), and $OD_{450}$ was read immediately using a SpectraMax M2 microplate reader (Molecular Devices). The cutoff for positive reactivity was 2 standard deviations above the mean OD of negative control wells coated with antigen but lacking serum.

## Flow cytometry and ICS assay

To generate single-cell lung leukocytes, lungs were cut into small pieces, digested with 1 mg/mL type I collagenase (Worthington) and 20 U/mL DNase I (Thermo Fisher Scientific) (30 min, 37 °C), mechanically dissociated using a gentleMACS Octo Dissociator, filtered through a 70-µm cell strainer, and red blood cells lysed with ACK lysing buffer (Gibco). To generate single-cell splenocytes, spleens were gently mashed with a syringe plunger, filtered through a 70-µm cell strainer, and treated with ACK lysing buffer.

Splenocytes or lung leukocytes were placed in 96-well round-bottom plates at $2×10^6$ cells/well in complete RPMI and stimulated with 10 µg/mL of SARS-CoV-2 peptides (37 °C). After 1 hour, Brefeldin A (BioLegend; 1:1000 dilution) and rat anti-mouse CD107a (Clone 1D4B, Biolegend) were added and the cells incubated for an additional 4 hours. Positive and negative controls were stimulated for the same time with Cell Stimulation Cocktail (eBioscience) or no stimulant, respectively. At the end of the 4-hour incubation, cells were stained with efluor 455 (UV) viability dye (Invitrogen) and fluorophore-conjugated antibodies against mouse CD3ε (Tonbo, clone 145-2C11), CD8α (BioLegend, clone 53-6.7), CD4 (eBioscience, clone GK1.5), CD44 (BioLegend, clone IM7), and CD62L (BioLegend, MFL-14). Cells were then fixed and permeabilized with Cytofix/Cytoperm and stained with fluorophore-conjugated antibodies against mouse IFNγ (Tonbo, clone XMG1.2), TNFα (eBioscience, clone MP6-XT22), IL-2 (BioLegend, clone JES6-5H4), CD107a (Biolegend, clone 1D4B), IL-4 (Biolegend, clone 11B11), and IL-17A (Biolegend, clone TC11-18H10.1). Data were collected on an LSR Fortessa flow cytometer (BD Biosciences) and analyzed using FlowJo software.

## IFNγ-ELISpot assay

Splenocytes or lung leukocytes, prepared as described above, were plated at $10^5$ cells/well in 96-well flat-bottom plates (Immobilon-P; Millipore, MA) pre-coated with anti-mouse IFNγ antibody (clone AN18; Mabtech, Sweden) and incubated (20 h, 37 °C) with 10 µg/mL of SARS-CoV-2 peptide. Plates were processed as previously described[147], and spot-forming cells (SFCs) counted with an ELISpot reader (MABTech).

## CD4+ T cell depletion

Mice were injected IP with CD4+ T cell-depleting antibody (BioXCell, clone GK1.5) or rat IgG2 isotype control antibody (BioXCell, clone LTF-2) (250 µg, IP) on days −3, −2, and −1 before SARS-CoV-2 challenge. Blood was collected before SARS-CoV-2 challenge from each mouse and analyzed by flow cytometry to validate CD4+ T cell depletion.

## Peptide prediction, selection, and immunization

HLA-DRB1*0101- and HLA-B*0702-restricted SARS-CoV2 N, S, and M protein-derived T cell epitopes were identified as follows. Protein sequences for SARS-CoV-2/human/USA/WA-CDC-WA1/2020 isolate (GenBank MN985325.1) were accessed via the NCBI protein database. Using the Immune Epitope Database (www.iedb.org) website tools and the "IEDB-recommended" method selection, MHC class II or class I peptide binding affinity predictions were obtained for all non-

redundant 15-mer peptides that bind the HLA-DRB1*0101 allele or for all 8- to 11-mer peptides that bind the HLA-B*0702 allele. The resulting peptide lists were sorted by increasing consensus percentile rank and the top 1% selected (Tables 1 and 2), synthesized, purified to ≥95% purity by TC Peptide Lab (San Diego) using reverse-phase HPLC, and validated by mass spectrometry. Peptides were dissolved in DMSO for use.

HLA-DRB1*0101- and HLA-B*0702-restricted SARS-CoV2 proteome-derived T cell epitopes were searched on the NIAID Virus Pathogen Database and Analysis Resource (https://www.viprbrc.org/; accessed May 2, 2021) by querying the virus species name "severe acute respiratory syndrome-related coronavirus" from "human" hosts. We limited our search to epitopes identified by at least 1 of the following T cell assays: ELISpot, ICS, or MHC-binding. The resulting 37 predicted HLA-DRB1*0101 and HLA-B*0702 epitopes (Tables 3 and 4) were synthesized as crude material (1 mg scale) by TC Peptide Lab (San Diego).

### Histopathology
Lungs were fixed with zinc formalin (24 h, room temperature) and transferred to 70% alcohol. Fixed lungs were embedded in paraffin using standard procedures, sliced into 4-μm sections, stained with H&E using a Leica ST5020 autostainer, and imaged with a Zeiss AxioScan Z1 (40×0.95 NA objective). Histopathological analysis was performed by a board-certified veterinary pathologist, who was blinded to group identity. Sections were scored (0–5) for 10 criteria for SARS-CoV-2-induced pneumonia, as seen in hamsters, macaques, and COVID-19 patients[148].

### Statistical analysis
Data are expressed as the mean ± standard error (SEM) and were analyzed with Prism software v9.1.1 (GraphPad Software). Differences between group means were analyzed by the Kruskal–Wallis test (>2 groups), nonparametric Mann–Whitney test (2 groups), 2-way ANOVA followed by Sidak's multiple comparisons test (>2 groups, >1 variable), or one-way ANOVA with Dunnett's post-hoc test. $P < .05$ was considered statistically significant.

### Reporting summary
Further information on research design is available in the Nature Portfolio Reporting Summary linked to this article.

## Data availability
All data supporting this study's findings are included in the text, figures, supplementary material, and the source data provided in this paper. The complete dataset of immunofluorescence and histopathology images has been deposited in the repository: https://doi.org/10.5281/zenodo.10397796[149]. Source data are provided in this paper.

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

## Acknowledgements

We thank the Department of Laboratory Animal Care (Morag Mackay, Pascual Barajas, and Joseph Garza), the Department of Environmental Health and Safety (Dr. Laurence Cagnon and David Hall), and Flow Cytometry Core (Cheryl Kim) at the La Jolla Institute for Immunology for their assistance. We also thank Dr. Alessandro Sette at the La Jolla Institute for Immunology for providing HLA-B*0702 and DRB1*0101 *Ifnar1*$^{-/-}$ breeder mice. This study was funded by National Institutes of Health grants U19 AI142790-02S1 (to E.O.S. and S.S.) and U01 AI149644 (to R.S.B.), the GHR Foundation (to S.S. and E.O.S.), the Arvin Gottleib Foundation (to S.S. and E.O.S), and the Overton family (to S.S. and E.O.S.).

## Author contributions

R.P.S.A., A.E.N., and S.S. conceived the study. R.P.S.A. and A.E.N. designed, performed, and analyzed the experiments. R.A., A.E.N., P.B.A.P., K.V., R.M., A.G., J.T., J.A.R.-N., M.N.N., E.M., S.L.-B., K.D., B.L., Z.M., S.M, N.S., and K.K. performed and analyzed experiments. S.R.L. and R.S.B. provided the MA10 virus strain. E.O.S. and S.S. obtained funding. S.S. supervised the research. R.P.S.A., A.E.N., and S.S. wrote the manuscript; other authors provided editorial comments.

## Competing interests

The authors declare no competing interests.

## Additional information

[1]Center for Infectious Disease and Vaccine Research, La Jolla Institute for Immunology, La Jolla, CA, USA. [2]Microscopy and Histology Core Facility, La Jolla Institute for Immunology, La Jolla, CA, USA. [3]Department of Epidemiology, University of North Carolina at Chapel Hill, Chapel Hill, NC, USA. [4]Histopathology Core Facility, La Jolla Institute for Immunology, La Jolla, CA, USA. [5]Department of Microbiology and Immunology, University of North Carolina at Chapel Hill, Chapel Hill, NC, USA. [6]Department of Medicine, Division of Infectious Diseases and Global Public Health, University of California, San Diego (UCSD), La Jolla, CA, USA. [7]Present address: Department of Microbiology and Pathology, University Center for Health Science (CUCS), University of Guadalajara, Guadalajara 44340, Mexico. ✉e-mail: aelong@lji.org; sujan@lji.org

