## [Peer Review File · Nature Communications]

REVIEWER COMMENTS

Reviewer #1 (Remarks to the Author):

The authors made good responses to the questions raised in the previous round of review. They once again highlighted the novelty of the study. Although the concept has been mentioned by some other studies, the community could also learn useful information that cross-reactive T cells could be induced by natural infection of a specific strain of common cold coronavirus OC43, or even vaccination of a T cell epitope peptide.

Some comments have not been well responded: as Specific comments 1) from R1, alignment of the peptides identified among all human coronaviruses should be performed to expand the significance; Specific comments 4) from R1, the absence of broadly binding/neutralizing antibodies following OC43 infection revealed by the current manuscript may due to the limitation of experiment method, and the epitope of cross-reactive T cells was concentrated on a limited number of peptides which means the frequency is also low.

Reviewer #2 (Remarks to the Author):

This manuscript provides direct evidence that Memory CD4 T cells elicited by common cold coronaviruses might promote enhanced SARS-CoV-2 control in HLA transgenic mice. Revisions made in response to reviewers' concerns are satisfactory and have enhanced the rigor of the paper. This reviewer has no further concerns with this manuscript.

Reviewer #3 (Remarks to the Author):

The authors have adequately addressed the queries raised in my previous review and i have no additional comments.

Reviewer #4 (Remarks to the Author):

This study makes two main claims:

Common cold virus OC43 infection elicits CD8+ and CD4+ effector T cells that cross-react with SARS-CoV-2 peptides.

There is a protective role for OC43-elicited SARS-CoV-2 cross-reactive CD4+ T cells.

Claim 1 provides some knowledge, but it's not entirely new information since previous studies (e.g., Mateus et al., 2020) in humans have demonstrated cross-reactivity between HCoVs and SARS-CoV-2 T-cell epitopes. The findings in Claim 2 are of high interest because there is evidence of cross-reactive protective immunity in humans. A mouse model may be appropriate to demonstrate the mechanism of protective immunity to SARS-CoV-2 acquired from previous exposure to HCoVs. Therefore, the authors utilized HLA-expressing transgenic mice to provide insight into cross-reactive protective immunity to SARS-CoV-2.

However, in its current form, this manuscript does not help advance our knowledge of the immune mechanisms underlying this cross-reactivity-mediated protection. These transgenic mice do not produce antibodies to the N-protein. While they do produce antibodies to the S-protein, they do not cross-react with the S protein of SARS-CoV-2 (Fig. S2). As seen in Fig. S4, CD8 T cells induced in response to OC43, and CD8-epitope immunization (Fig 5), do not have a significant impact on the SARS-CoV-2 challenge in terms of both viral RNA and pathology. Although CD4 depletion experiments in OC43-infected mice showed an increase in viral RNA and higher N-protein expression post SARS-CoV-2 challenge, the pathology is not different between CD4-depleted and control mice. The authors proposed that the protection may be CD8-mediated, but unfortunately, they have decided to remove the data on CD8-depletion experiments.

Altogether, it appears that none of these three immune mechanisms (CD4, CD8, and antibodies) seem to drive protective immunity during this cross-reactive immune protection phenomenon. The question then arises: what else is protecting the OC-43-infected mice from the SARS-CoV-2 challenge (Fig. 6)? Without a clear indication or mechanistic insight into the cross-reactive immunity-mediated protection against SARS-CoV-2, the study does not address the original question for which it was designed in the transgenic animal models. I believe that if the model is appropriate, it presents a great opportunity for the authors to decipher how cross-reactive immunity from previous exposure to a common cold virus confers protection against SARS-CoV-2 infection or disease. Since the authors are proposing that their study will advance efforts for a broadly protective T-cell epitope-based vaccine, it becomes even more important to address how T cells, CD4 or CD8 or both, are actually implicated in this protection acquired from previous exposure to HCoVs.

REVIEWER COMMENTS

Reviewer #1 (Remarks to the Author):

The authors made good responses to the questions raised in the previous round of review. They once again highlighted the novelty of the study. Although the concept has been mentioned by some other studies, the community could also learn useful information that cross-reactive T cells could be induced by natural infection of a specific strain of common cold coronavirus OC43, or even vaccination of a T cell epitope peptide.

Some comments have not been well responded: as Specific comments 1) from R1, alignment of the peptides identified among all human coronaviruses should be performed to expand the significance; Specific comments 4) from R1, the absence of broadly binding/neutralizing antibodies following OC43 infection revealed by the current manuscript may due to the limitation of experiment method, and the epitope of cross-reactive T cells was concentrated on a limited number of peptides which means the frequency is also low.

We thank the reviewer for their positive comments and appreciate their suggestions towards improving the significance of our findings.

Per the reviewer's suggestion, we conducted an extensive sequence alignment analysis to assess the conservation of the identified epitopes across various human coronaviruses (shown in Figure 1 below; not included in the revised manuscript). The analysis revealed percentage identities of the individual epitopes between strains of 0–40% in NL63, 0–38.46% in 229E, 0–46.67% in HKU1, and 16–40% in OC43. Based on this analysis and our study design (which was focused on identifying OC43-elicited T cell responses that are cross-reactive with SARS-CoV-2), we consider it unlikely that the OC43/SARS-CoV-2-cross-reactive epitopes identified in our study would exhibit high enough similarity with other common cold coronaviruses. Further, this analysis suggests that future research should examine how prior NL63, 229E, or HKU1 exposure influences T cell responses to SARS-CoV-2 by developing HLA transgenic mouse models of sequential infections with NL63, 229E, or HKU1 followed by SARS-CoV-2. We now mention the need to develop these new mouse models (lines 416-423).

We also appreciate the reviewer's insightful comments regarding the absence of broadly binding/neutralizing Abs following OC43 infection and the limited frequency of cross-reactive T cell epitopes identified in our study. These points are valid and now acknowledged in the revised manuscript within the study limitations section (lines 406–426). As noted, our experimental approach has inherent limitations, including the inability to capture all potential Ab and T cell epitopes comprehensively. We have emphasized the

need for further research to expand upon our findings and explore a wider spectrum of potential cross-reactive epitopes.

Reviewer #2 (Remarks to the Author):

This manuscript provides direct evidence that Memory CD4 T cells elicited by common cold coronaviruses might promote enhanced SARS-CoV-2 control in HLA transgenic mice. Revisions made in response to reviewers' concerns are satisfactory and have enhanced the rigor of the paper. This reviewer has no further concerns with this manuscript.

We thank the reviewer for their comments.

Reviewer #3 (Remarks to the Author):

The authors have adequately addressed the queries raised in my previous review and I have no additional comments.

We thank the reviewer for their comments.

Reviewer #4 (Remarks to the Author):

This study makes two main claims:

Common cold virus OC43 infection elicits CD8+ and CD4+ effector T cells that cross-react with SARS-CoV-2 peptides.

There is a protective role for OC43-elicited SARS-CoV-2 cross-reactive CD4+ T cells.

Claim 1 provides some knowledge, but it's not entirely new information since previous studies (e.g., Mateus et al., 2020) in humans have demonstrated cross-reactivity between HCoVs and SARS-CoV-2 T-cell epitopes. The findings in Claim 2 are of high interest because there is evidence of cross-reactive protective immunity in humans. A mouse model may be appropriate to demonstrate the mechanism of protective immunity to SARS-CoV-2 acquired from previous exposure to HCoVs. Therefore, the authors utilized HLA-expressing transgenic mice to provide insight into cross-reactive protective immunity to SARS-CoV-2.

However, in its current form, this manuscript does not help advance our knowledge of the immune mechanisms underlying this cross-reactivity-mediated protection. These transgenic mice do not produce antibodies to the N-protein. While they do produce antibodies to the S-protein, they do not cross-react with the S protein of SARS-CoV-2 (Fig. S2). As seen in Fig. S4, CD8 T cells induced in response to OC43, and CD8-

epitope immunization (Fig 5), do not have a significant impact on the SARS-CoV-2 challenge in terms of both viral RNA and pathology. Although CD4 depletion experiments in OC43-infected mice showed an increase in viral RNA and higher N-protein expression post SARS-CoV-2 challenge, the pathology is not different between CD4-depleted and control mice. The authors proposed that the protection may be CD8-mediated, but unfortunately, they have decided to remove the data on CD8-depletion experiments.

Altogether, it appears that none of these three immune mechanisms (CD4, CD8, and antibodies) seem to drive protective immunity during this cross-reactive immune protection phenomenon. The question then arises: what else is protecting the OC-43-infected mice from the SARS-CoV-2 challenge (Fig. 6)? Without a clear indication or mechanistic insight into the cross-reactive immunity-mediated protection against SARS-CoV-2, the study does not address the original question for which it was designed in the transgenic animal models. I believe that if the model is appropriate, it presents a great opportunity for the authors to decipher how cross-reactive immunity from previous exposure to a common cold virus confers protection against SARS-CoV-2 infection or disease. Since the authors are proposing that their study will advance efforts for a broadly protective T-cell epitope-based vaccine, it becomes even more important to address how T cells, CD4 or CD8 or both, are actually implicated in this protection acquired from previous exposure to HCoVs.

We thank the reviewer for this detailed feedback, which has helped us to understand their specific concerns. We have sought to address them as follows:

We agree that the cross-reactivity between common cold HCoVs and SARS-CoV-2 has been the subject of numerous prior studies in humans. However, those studies also had several limitations, including sparse details regarding the precise infecting HCoVs and the times of infection. Our research aimed to elucidate the protective role of OC43-cross-reactivity in distinct but tightly controlled contexts that allowed us to evaluate the quantitative and qualitative contributions of OC43 pre-exposure to subsequent infection with SARS-CoV-2. We fully recognize that the specificity of our study is also one of its limitations and have incorporated discussions with respect to the real-world scenario in the limitations section (lines 406-426).

In particular, we note that:

- Our transgenic mouse strains expressed a single transgenic HLA class I or II molecule but not both, which undoubtedly influenced the responses. As an example, SARS-CoV-2 RNA levels in the lung were lower in HLA-DRB1*0101 *Ifnar1*^{-/-} mice (Figure 6B – OC43 group mean: 2.91) compared with HLA-B*0702 *Ifnar1*^{-/-} mice at day 16 post-OC43 infection (supplementary Figure 4C – OC43 group mean: 7.27), and a comparably low RNA level was only seen in the HLA-B*0702 *Ifnar1*^{-/-} mice at 60 days post-infection (Figure 5H – OC43 group mean: 3.63). As shown in Supplementary Figures S2B and S2C, strain-specific differences in RNA persistence and distribution exist, potentially explaining the observed differences in viral load between the mouse strains.

- Contrary to our initial expectations that cross-protective immunity may result from a single mechanism, our data implicate multiple concurrent mechanisms. Depending on the context, this multifaceted protection likely involves CD4 and CD8 T cells and Abs to varying extents. Although cross-reactive anti-OC43 and anti-SARS-CoV-2 Abs have been identified, the cross-reactivity is primarily targeted against the S2 protein, which is a target for non-neutralizing Abs (see Crowley A et al. ¹). This suggests that Abs are not likely to be the principal protective mediators.

- We also note that our study was restricted to evaluation at day 3 post-infection. This time frame was chosen to observe the cross-reactive memory response (before the primary T cell response to SARS-CoV-2 is detectable); however, it may not be optimal for a comprehensive assessment of lung pathology.

Moving forward, we anticipate that more exhaustive explorations of the possible protective mechanisms of cross-reactivity and a deeper understanding of the effects on lung pathology will be valuable. We appreciate the insights provided by the reviewers and are committed to advancing our understanding in this area.

1. Crowley, A.R. *et al.* Boosting of cross-reactive antibodies to endemic coronaviruses by SARS-CoV-2 infection but not vaccination with stabilized spike. *Elife* **11** (2022).

A

Peptide LFLFFSNVTFHAI (S₅₄₋₆₈)

SARS-CoV-1 S SSMRGVYYP-DEIFRSDTLYLTLQ**LFLFFYSNV--TGFH**-----TIN-----HTFG 77
 SARS-CoV-2 S SFTRGVYYP-DKVFRRSSVLHSTQ**LFLFFFSNV--TFWH**-----AIHVSGTNG--TKRFD 80
 MERS S SKADGIIYPOGRTYNSI-----T**ITYOGLFFYQGDHGD**MYVYSAGHATGTFP--QKLFV 102
 HKU1 S SYGLGTYYLLDRVYLNT-----T**ILFTGYFFPKSGANFR**-----DLSLKGTYYLSTLWYQ 91
 NL63 S YSANGFFYIDVGNHRS-----A**ALHTGYDYDA--NQYY**IYVTNEIGLNASVTLKICKFS 109
 229E S -----
 OC43 S TNGLTYYVLLDRVYLNT-----**LFLNGYYPTSGSTYR**-----NMALKGSVLLSRLWFK 92

Peptide AYYVGYLQPRTELLK (S₂₆₄₋₂₇₈)

SARS-CoV-1 S ILTAFS-----P---AQDIWGTSA**AYYVGYLQPRTEMLK**DE 268
 SARS-CoV-2 S LLALHRSYL-----TPGD---SSSGWTAGAA**AYYVGYLQPRTELLK**NE 281
 MERS S IPHSIR-----SIQS---DRKAW---**AYYVGYLQPRTELLK**SV 329
 HKU1 S LPLTCN-----A---ISSNTDNET**QYVVTPLSKRQYLLK**DN 272
 NL63 S LPPTVREIVVARTGQGFYINGFKYFDLGFTEAVNFNVTTASA**DFWTVAFATFVDVLI**VVS 428
 229E S LPKTVREFVISRTGHFYINGRYFTLGVTEAVNFNVTTAET**DFCTVALASYADLV**SVS 245
 OC43 S MPLTCN-----S-----KLT**EYVVTPLTSRQYLLA**NQ 280

Peptide SFCTQLNRALTGIAV (S₇₅₈₋₇₇₂)

SARS-CoV-1 S QYGS**FC**TQLNRAL**SLAA**EQDRNTRVFAQVKQMYKTPTLKYFG----GF-NFSQILP- 789
 SARS-CoV-2 S QYGS**FC**TQLNRAL**SLAA**EQDKNTOEVFAQVKQIYKTPPIKDFG----GF-NFSQILP- 807
 MERS S EYGG**FC**SKIN**QALHG**ANLRQDDSVRNLFASFVSSQSSPIIPGFG----GDFNLLTLEP- 876
 HKU1 S EYGT**FC**DNIN**SLDE**VNGLDDTTQLHVADTLMQGVTLSSNLTNLHFVDNINFKSLVGC 894
 NL63 S QYTS**ACT**I**EDAL**R**LSA**HLETNVSSMLTFDSNAFSLANVTSF----GDYNLSSVLFQ 859
 229E S QYTS**ACT**I**EDAL**R**NSA**HEADVSEMLTFDKKAPTLANVSSF----GDYNLSSVLF 678
 OC43 S EYGS**FC**DNIN**AL**TEVNELDTTQLQVANSMLMNGVTLSTKLKDGVNFNVDNINFSVFLGC 892

Peptide YRRATRIRRGDGK (N₈₆₋₁₀₀)

SARS-CoV-1 N QGLEPNTASWFTALQHGK-EELRFPRGQVPIINTNSGPDQIQ**YRRATRIR--VRGGDGK** 101
 SARS-CoV-2 N QGLEPNTASWFTALQHGK-EDLKFPFGQVPIINTNSGPDQIQ**YRRATRIR--IRGGDGK** 100
 MERS N RAAPNNTSWYTGTLQHGK-VPLTFPPGQGVPLNANSTPAQNA**YWRRODRK--INTGNG** 90
 HKU1 N QGNTI PHYSWFSGITQFQKGRDFKFSDDGQVPIAFGVPSPSEAK**YWRHRSRSFKTADGQ** 114
 NL63 N ---KFPFSPFYMLLWSSDKAPYRVI PRNLVPTIGKGN-KDEQIQ**YWNVQER--WRMRGQ** 68
 229E N ---GRIPYSLSYSLVLDSE-QQFQWVTPRNLVPTMKKD-KNKLQ**YWNVQER--FRMRGK** 70
 OC43 N GGNVVPPYYSWFSGITQFQKGEFEFVEGQVPIAPGVPAEAK**YWRHNRSEFKTADGN** 115

Peptide DAALALLLDRINQL (N₂₁₆₋₂₃₀)

SARS-CoV-1 N SGGQ**DA**LALLLDRINQLQSK-VSGKG--Q-QQQGQTVT-----KKSAA--- 253
 SARS-CoV-2 N GNGQ**DA**LALLLDRINQLQSK-MSGKG--Q-QQQGQTVT-----KKSAA--- 252
 MERS N SGIQ**AV**GGDLLYLDLNLRLQAL-ESGKV--K-QSQPKVIT-----KKDAA--- 244
 HKU1 N ----**DS**IVK**PDMA**DEIENL-VLAKLGKD-S-KFQQVTV-----KQNAKEIR 261
 NL63 N DLVA**AV**TALAK**NLG**EDNQSKSPSS--GTSPEKFNKFL-----SQPRADK 227
 229E N DIM**AV**A**ALK**SL**IG**EDKPKKDKKSAKTGPKPSRNQSPASSQTSAKSLARSQSSETEKQ 240
 OC43 N ----**TS**GV**TP**MD**AD**QIASL-VLAKLGKD-ATKQQVTV-----KHTAKEVR 263

Peptide ICLLQFAYANRRFL (M₃₂₋₄₆)

SARS-CoV-1 M -----MADNGTITVEELKQLLEQWNLVIGFLEFL**IMLLQFAYSNRRFL**IIKLVFLW 54
 SARS-CoV-2 M -----MADNGTITVEELKQLLEQWNLVIGFLEFL**ICLLQFAYANRRFL**IIKLIPLW 55
 MERS M -----MSNMTQLTEAQIIAIIKDWNFAWSLIFLL**ITIVLQGYPSRSMIV**VFKRMFLW 54
 HKU1 M -----MNKSLFQFTSDQAVTFKKEWNFSLGVILLE**ITILQGYTSRSMFV**LIKMIILW 56
 NL63 M ----**MS**NSV**PL**LEV**V**ILRNW**NS**WNLIL**TL****ITV**L**V**L**Q**GH**Y**K**Y**S**R**L**L**GLK**M**L**V**LW 53
 229E M ----**MS**ND--**NC**T**GD**IV**TH**L**K**W**N**F**GN**V**LT****ITV**L**V**L**Q**GH**Y**K**Y**S**R**L**L**GLK**M**L**V**LW 52
 OC43 M MSSKTTAPVYIWTADDAIKFLKEWNFSLGIILL**ITILQGYTSRSMFV**VIKMIILW 60

Peptide VLAAYRINWITGGI (M₆₆₋₈₀)

SARS-CoV-1 M LLWVPTLAC**VLA**AV--**YR**INWIT**GGI**AIAMACIVGLMWLSYFVASFRLFARTSRMWSFN 112
 SARS-CoV-2 M LLWVPTLAC**VLA**AV--**YR**INWIT**GGI**AIAMACLVGLMWLSYFIASFRLFARTSRMWSFN 113
 MERS M LLWPSSMAL**SIF**SAV--**Y**PID**IA**S**Q**ITSGIVAASVAMMWISYFVQSIRLFEMRTGSGWWSFN 112
 HKU1 M LMWPLTITL**IENC**V--**Y**AL**NN**A**L**A**P**SIVFTTISIVWILYFVNSIRLFRITGSGWWSFN 114
 NL63 M CLWPLVLAL**I**ED**C**F**V**IN**V**D**V**F**FG**PIILMSITTLCLWVMYFVNSIRLFRWRVKTFWAFN 113
 229E M LLWPLVLAL**I**ED**T**WAN**DS**N**WA**F**V**A**P**SIFMAVSTLVMMVMYFANSIRLFERRARTFWAFN 112
 OC43 M LMWPLTITL**I**ENC**V**--**Y**AL**NN**V**L**GLSIVFTTIVAIMMWIVYFVNSIRLFRITGSGWWSFN 118

B

Figure 1. Mapping the HLA-DRB1*0101-restricted epitopes elicited by OC43 that cross-react with SARS-CoV-2 across human coronaviruses. (A) Multiple sequence alignment of S, N, and M proteins from human coronaviruses: SARS-CoV-1 (AY278741.1), SARS-CoV-2 (NC_045512.2), MERS (NC_019843.3), HKU1 (NC_006577.2), NL63 (NC_005831.2), 229E (NC_002645.1), and OC43 (NC_006213.1). The putative locations of identified cross-reactive epitopes are boxed, and sequences are shown in bold. Dashes indicate deletions. **(B)** Percent identity of SARS-CoV-2 cross-reactive peptides and putative epitopes in human common cold coronaviruses.

REVIEWERS' COMMENTS

Reviewer #1 (Remarks to the Author):

The quality of this manuscript has not been significantly improved. Here are some concerns:

1. As mentioned by the authors and some reviewers, the N104-113 peptide identified and studied in this manuscript has been described and discussed in several published studies. Although the authors explore this epitope in HLA-B*0702 and HLA-DRB1*0101 Ifnar^{-/-} transgenic mouse models, there is still a lack of innovative concepts.
2. The current study focuses on the CD4 T cell responses while the authors somehow neglect discussing the subsequent cross antibodies responses. The IgG titer against OC43 spike may be quite low which results in the undetectable anti-SARS-CoV-2 spike titer.
3. Since there are 4 common cold coronaviruses, while the authors only studied OC43, the title of the current manuscript should narrow the range to common cold coronavirus OC43.
4. The HLA-DRB1*0101-restricted epitopes elicited by OC43 that cross react with SARS-CoV-2 across human coronaviruses identified in this study share very limit amino acid sequences between OC43 and SARS-CoV-2, which is raises another important issue to be discussed.

Reviewer #1 (Remarks to the Author):

The quality of this manuscript has not been significantly improved. Here are some concerns:

1. As mentioned by the authors and some reviewers, the N104-113 peptide identified and studied in this manuscript has been described and discussed in several published studies. Although the authors explore this epitope in HLA-B*0702 and HLA-DRB1*0101 *Ifnar*^{-/-} transgenic mouse models, there is still a lack of innovative concepts.

We appreciate the reviewer's comment. We agree that the N104-113 peptide recognized by CD8⁺ T cells has been identified by several studies (as cited in the discussion lines 390-397). However, the precise infection parameters (in terms of the identity, number, sequence, and timing of infections with the various common cold HCoV_s) are unknown in the published human studies, and importantly, human studies have provided associative data suggesting that common cold HCoV/SARS-CoV-2 cross-reactive T cells associate with protection. In contrast, our study shows that a single pre-exposure with OC43 elicits CD8 T cells that are cross-reactive with SARS-CoV-2, and provides direct evidence that the pre-existing OC43-elicited T cells contribute to cross-protection against subsequent infection with SARS-CoV-2.

2. The current study focuses on the CD4 T cell responses while the authors somehow neglect discussing the subsequent cross antibodies responses. The IgG titer against OC43 spike may be quite low which results in the undetectable anti-SARS-CoV-2 spike titer.

We agree that OC43-induced antibody response in our mouse model is weak. This is likely because OC43 does not establish sufficient levels of infection to induce robust antibody responses, whereas OC43 is able to elicit detectable SARS-CoV-2-cross-reactive CD4⁺ T cell responses in our mouse model. We therefore focused the present study on evaluating the protective function of these cross-reactive CD4⁺ T cells. Our results are in line with human studies reporting that the cross-reactive antibody response is weak and decays rapidly, and that the cross-reactive cellular immunity (ie, not antibodies) likely contributes to protection against SARS-CoV-2^{1, 2, 3, 4, 5}. We have referenced these published studies in the discussion (lines 372, and 373-374).

3. Since there are 4 common cold coronaviruses, while the authors only studied OC43, the title of the current manuscript should narrow the range to common cold coronavirus OC43.

We thank the reviewer for this suggestion, and agree that the title should be changed. The new title is: Human coronavirus OC43-elicited CD4⁺ T cells protect against SARS-CoV-2 in HLA-DRB1*0101 transgenic mice.

4. The HLA-DRB1*0101-restricted epitopes elicited by OC43 that cross react with SARS-CoV-2 across human coronaviruses identified in this study share very limit amino acid sequences between OC43 and SARS-CoV-2, which is raises another important issue to be discussed.

We acknowledge the reviewer's observation that some SARS-CoV-2 cross-reactive peptides in our study show limited homology with the OC43 sequence. However, this is not surprising given the low cytokine production (IFN_γ alone or IFN_γ and TNF) in response to some epitopes. Our results are also consistent with human data: variable but low CD4⁺ T cell cross-reactivity to SARS-CoV-2 is present in many healthy SARS-CoV-2-unexposed individuals, and the degree of amino acid sequence identity between common cold HCoV_s and SARS-CoV-2 does not correlate with CD4⁺ T cell cross-reactivity⁶. We have included these points in the discussion (lines 376-381).

1. Anderson EM, *et al.* Seasonal human coronavirus antibodies are boosted upon SARS-CoV-2 infection but not associated with protection. *Cell* **184**, 1858-1864 e1810 (2021).
2. Ng KW, *et al.* Preexisting and de novo humoral immunity to SARS-CoV-2 in humans. *Science (New York, NY)* **370**, 1339-1343 (2020).
3. Poston D, *et al.* Absence of SARS-CoV-2 neutralizing activity in pre-pandemic sera from individuals with recent seasonal coronavirus infection. *medRxiv : the preprint server for health sciences*, (2020).
4. Premkumar L, *et al.* The receptor binding domain of the viral spike protein is an immunodominant and highly specific target of antibodies in SARS-CoV-2 patients. *Science immunology* **5**, (2020).
5. Mateus J, *et al.* Selective and cross-reactive SARS-CoV-2 T cell epitopes in unexposed humans. *Science (New York, NY)* **370**, 89-94 (2020).
6. Loyal L, *et al.* Cross-reactive CD4(+) T cells enhance SARS-CoV-2 immune responses upon infection and vaccination. *Science (New York, NY)* **374**, eabh1823 (2021).